# Symbolic Learning to Optimize: Towards Interpretability and Scalability

**Wenqing Zheng**[1]**, Tianlong Chen**[1]**, Ting-Kuei Hu**[2]**, Zhangyang Wang**[1]
[1]The University of Texas at Austin, [2]Texas A&M University
{w.zheng,tianlong.chen,atlaswang}@utexas.edu, tkhu@tamu.edu

## Abstract

Recent studies on *Learning to Optimize* (**L2O**) suggest a promising path to automating and accelerating the optimization procedure for complicated tasks. Existing L2O models parameterize optimization rules by neural networks, and learn those *numerical* rules via meta-training. However, they face two common pitfalls: (1) *scalability*: the numerical rules represented by neural networks create extra memory overhead for applying L2O models, and limit their applicability to optimizing larger tasks; (2) *interpretability*: it is unclear what an L2O model has learned in its black-box optimization rule, nor is it straightforward to compare different L2O models in an explainable way. To avoid both pitfalls, this paper proves the concept that we can "kill two birds by one stone", by introducing the powerful tool of *symbolic regression* to L2O. In this paper, we establish a holistic symbolic representation and analysis framework for L2O, which yields a series of insights for learnable optimizers. Leveraging our findings, we further propose a lightweight L2O model that can be meta-trained on large-scale problems and outperformed human-designed and tuned optimizers. Our work is set to supply a brand-new perspective to L2O research. Codes are available at: https://github.com/VITA-Group/Symbolic-Learning-To-Optimize.

## 1 Introduction

Efficient and scalable optimization algorithms (a.k.a., *optimizers*) are the cornerstone of almost all computational fields. In many practical applications, we will repeatedly perform a certain type of *optimization tasks* over a specific distribution of data. Each time, the inputs that define the optimization problem are new but similar to previous related tasks. We say such an application has a narrow *task distribution*. Conventional optimizers are theory-driven, so they obtain performance guarantees over the classes of problems specified by the theory. Instead of manually designing and tuning task-specific optimizers, one may be naturally interested to pursue more general-purpose and data-driven optimization frameworks.

Learning to optimize (**L2O**) (Almeida et al., 2021; Andrychowicz et al., 2016; Chen et al., 2017; Li & Malik, 2017a) is an emerging paradigm that automatically develops an optimizer by learning from its performance on a set of optimization tasks. This data-driven procedure generates optimizers that efficiently solve problems similar to those in the training tasks. Although often lacking a solid theoretical basis, the training process shapes the *learned optimizer* according to problems of interest. When the task distribution is concentrated, the learned optimizer can specifically fit the tasks by discovering "shortcuts" that classic optimizers do not take (Li & Malik, 2017a;b). L2O has demonstrated many practical benefits including faster convergence and/or better solution quality (Chen et al., 2021a), even saving clock time (Li et al., 2020b) and hardware energy (Li et al., 2020a).

With promising progress being made, the questions persist: *how far is L2O from becoming principled science? To what extent can we trust L2O as being mature for real applications?* As a fast-growing new field, fascinating open challenges remain concerning L2O, among which two particular barriers will be focused on in this work:

- **Interpretability.** The learned optimizers are often *hard to interpret*. Traditional algorithm design follows analytic principles and admits "white box" symbolic forms, which are clear

to explain, analyze, troubleshoot, and prove. In contrast, existing L2O approaches fit update models using sophisticated data-driven predictors such as neural networks. As a result, it is unclear what rules have been learned in those "black boxes", prohibiting their further explainable analysis and comparison.

- **Scalability.** The learned optimizers can be *hard to scale*. L2O methods use learnable predictors to fit update rules, which incur extra overhead in memory that grows proportionally with the amount of optimization task variables. Moreover, the training of a learned optimizer often involves backpropagating through a full unrolled sequence for the parameter updates in the optimization tasks, and the memory cost grows fast with the unroll length. The scale of that growth results in complexities that current L2O methods cannot handle.

Our proposed solution, centered at addressing the two research gaps, is a brand-new **symbolic representation and interpretation framework for L2O**. While benefiting from large modeling capacity, L2O update rules fit by numerical predictors are inexplainable and memory-heavy. In contrast, the symbolic representations of traditional optimizers enjoy low memory overhead and clear explainability. We aim to integrate the strengths of symbolic forms into L2O. We propose to convert a numerical L2O predictor into a symbolic form that preserves the same problem-specific performance and is still tunable. Those symbolic rules are light, scalable, and interpretable. They can even be made tunable and subject to further meta testing, by a lightweight trainable re-parameterization. Based on them, we will further develop a set of L2O interpretability metrics.

Our main **technical innovations** are summarized below:

- We build the first *symbolic learning to optimize* (**symbolic L2O**) framework that is generally applicable to existing L2O approaches. That includes defining the optimizer symbolic space and the fitting approach to convert a numeric L2O to its symbolic counterpart.

- Under the symbolic L2O framework, we establish a unified scheme for L2O interpretability. Our synergistic analysis in the symbolic domain uncovers a number of new insights that are otherwise difficult to observe, including what "shortcuts" L2O methods essentially learn and how they differs across various methods.

- The lightweight symbolic representation also allows L2O to scale better, and we introduce a lightweight trainable re-parameterization to enable further fine-tuning of symbolic rules. **For the first time**, we obtain a symbolic optimizer that can be meta-trained on ResNet-50 level (23.5 million parameters) optimizee within 30 GB GPU memory, and achieved the state-of-the-art (SOTA) performance when applied to training larger (ResNet-152, 58.2 million parameters) or very different deep models (MobileNet v2 (Sandler et al., 2018)) on various datasets, outperforming existing manually designed and tuned optimizers.

## 2 RELATED WORKS

L2O lies at the intersection of ML and optimization research. It overlaps with meta-learning (Vilalta & Drissi, 2002; Hospedales et al., 2021), which contributes a significant portion of L2O development (Andrychowicz et al., 2016; Li & Malik, 2017a). L2O captures two main aspects of meta learning: rapid learning within each task, and transferable learning across many similar tasks. Using data-driven *predictors* to fit update rules, recent L2O approaches have shown success for a wide variety of problems, from convex optimization (Gregor & LeCun, 2010), nonconvex optimization (Andrychowicz et al., 2016) and minmax optimization (Shen et al., 2021), to black-box optimization (Chen et al., 2017) and combinatorial optimization (Khalil et al., 2017). Application areas benefiting the most from L2O methods include signal processing and communication (Borgerding et al., 2017; Balatsoukas-Stimming & Studer, 2019; You et al., 2020), image processing (Zhang & Ghanem, 2018; Corbineau et al., 2019; Chen et al., 2020b), medical imaging (Liang et al., 2020; Yin et al., 2021) and computational biology (Cao et al., 2019; Chen et al., 2019).

The pioneering L2O work (Andrychowicz et al., 2016) leverages a long short-term memory (LSTM) as the coordinate-wise predictor, which is fed with the optimizee gradients and outputs the optimizee parameter updates. Chen et al. (2017) takes the optimizee's objective value history as the input state of a reinforcement learning agent, which outputs the updates as actions. To enhance L2O generalization, (Lv et al., 2017) proposes random scaling and convex function regularizers tricks. (Wichrowska et al.,

2017) introduces a hierarchical RNN to capture the relationship across the optimizee parameters and trains it via meta-learning on the ensemble of small representative problems.

Training LSTM-based L2O models is however notoriously difficult, that is rooted in the LSTM-style unrolling during L2O meta-training (Tallec & Ollivier, 2017; Metz et al., 2018; Pascanu et al., 2013; Parmas et al., 2019). A practical optimizer may need to take thousands or more of iterations. However, naively unrolling LSTM to this full length is impractical for both memory cost and trainability by gradient propagation. Most LSTM-based L2O methods (Andrychowicz et al., 2016; Chen et al., 2017; Lv et al., 2017; Wichrowska et al., 2017; Metz et al., 2018; Cao et al., 2019) hence take advantage of truncating the unrolled optimization, yet at the price of the so-called "truncation bias" (Lv et al., 2017) that hampers their learned optimizer's generalization to unseen tasks (Chen et al., 2020a).

Most L2O approaches are "black boxes" with neither theoretical backup nor clear explanation on what they learn. A handful of efforts have been made on "demystifying" L2O, mainly about linking or reducing their behaviors to those of some analytic, better understood optimization algorithms. Maheswaranathan et al. (2020) analyzes and visualizes the learned optimizers, to discover that they have learned canonical mechanisms including momentum, gradient clipping, learning rate schedules, etc. Another well-known success was on interpreting the learned sparse coding algorithm (Gregor & LeCun, 2010), as related to a specific matrix factorization of the dictionary's Gram matrix (Moreau & Bruna, 2017), to the projected gradient descent trade-off between convergence speed and reconstruction accuracy (Giryes et al., 2018), or simply reduced to the original iterative algorithm with data-driven hyperparameter estimation (Chen et al., 2018; Liu et al., 2019; Chen et al., 2021b). Other examples include analyzing the learned alternating minimization on graph recovery (Shrivastava et al., 2020), and an analogy between deep-unfolded gradient descent and Chebyshev step-sizes (Takabe & Wadayama, 2020). However, they are all restricted to some specific objective or algorithm type.

## 3 TECHNICAL APPROACH

### 3.1 NOTATION AND PRELIMINARIES

Without loss of generality, let us consider an optimization problem $\min_{\mathbf{x}} f(\mathbf{x})$ where $\mathbf{x} \in \mathbb{R}^d$. Here, $f(\mathbf{x})$ is termed the optimization problem or *optimizee*, $\mathbf{x}$ is the variable to be optimized, and the algorithm to solve this problem is termed the *optimizer*. A classic optimizer usually iteratively updates $\mathbf{x}$ based on a handcrafted rule. For example, the first-order gradient descent algorithm takes an update at iteration $t$ based on the local gradient at the instantaneous point $\mathbf{x}_t : \mathbf{x}_{t+1} = \mathbf{x}_t - \alpha \nabla f(\mathbf{x}_t)$, where $\alpha$ is the step size. As a new learnable optimizer, L2O has much greater room to find flexible update rules. We define the input of L2O as $\mathbf{z}_t$. In general, $\mathbf{z}_t$ contains the optimization historical information available at time $t$, such as the current/past iterates $\mathbf{x}_0, \ldots, \mathbf{x}_t$, and/or their gradients $\nabla f(x_0), \ldots, \nabla f(\mathbf{x}_t)$, etc. L2O models an update rule by a *predictor* function $g$ of $\mathbf{z}_t$: $\mathbf{x}_{t+1} = \mathbf{x}_t - g(\mathbf{z}_t, \phi)$, where the mapping of $g$ is parameterized by $\phi$. Finding an optimal update rule can be formulated mathematically as searching for a good $\phi$ over the parameter space of $g$. Practically, $g$ is often a neural network (NN). Since NNs are universal approximators (Hornik et al., 1989), L2O has the potential to discover completely new update rules without relying on existing rules.

In order to find a desired $\phi$ associated with a fast optimizer, (Andrychowicz et al., 2016) proposed to minimize the weighted sum of the objective function $f(\mathbf{x}_t)$ over a time span $T$:

$$\min_{\phi} \mathbb{E}_{f \in \mathcal{T}} \left[ \sum_{t=0}^{T-1} w_t f(\mathbf{x}_t) \right], \quad \text{with} \quad \mathbf{x}_{t+1} = \mathbf{x}_t - g(\mathbf{z}_t, \phi), \ t = 0, \ldots, T-1 \quad (1)$$

where $w_0, \ldots, w_{T-1}$ are the weights whose choices depend on empirical settings. $T$ is also called the unrolling length. $f$ represents a sample optimization task from an ensemble $\mathcal{T}$ that represent the target task distribution. Note that $\phi$ determines the objective value through determining the iterates $\mathbf{x}_t$. L2O solves the problem (1) for a desirable $\phi$ and correspondingly the update rule $g(\mathbf{z}_t, \phi)$.

A typical L2O workflow is divided into two stages (Chen et al., 2021a): a *meta-training* stage that learns the optimizer with a set of similar optimization tasks from the task distribution; and a *meta-testing* stage that applies the learned optimizer to new unseen optimization tasks. The meta-training process often occurs offline and is time consuming. However, the online application of the method at meta-testing is (aimed to be) time saving.

### 3.2 MOTIVATION AND CHALLENGES

If we take a unified mathematical perspective, the execution of an optimizer can be represented either through a *symbolic* rule, or by *numeric* computation. Current L2O approaches parameterize their update rules $g$ by *numerical predictors* that can be learned from data via meta-training. They often choose sophisticated predictive models such as LTSMs (Andrychowicz et al., 2016; Lv et al., 2017; Chen et al., 2020a; Wichrowska et al., 2017), that predict the next update based on the current and historical optimization variables.

Despite the blessing of large learning capacity, the NN-based update rules are *uninterpretable* "black boxes". It is never well understood what rules existing L2O models have learned; nor is it easy to compare different L2O models and assess which has discovered more advanced rules. Moreover, numerical predictors, especially RNN-based ones, limit L2O scalability through severe *memory bottlenecks* (Metz et al., 2019; Wu et al., 2018). Their meta-training involves backpropagating through a full unrolled sequence of length $T$, for the parameter updates in their optimization task. At meta-testing time (though requiring less memory than meta-training), one has to store RNN's own parameters $\phi$ and hidden states, which are still far from negligible.

Those important gaps urge us to look back at symbolic update rules. Almost all traditional optimizers can be seen as formula-based symbolic methods within our framework, such as stochastic gradient descent (SGD), Adam, RMSprop, and so on. Their forms hinge on manual discovery and expert crafting. A symbolic optimizer has little memory overhead, and can be easily interpreted or transferred like an equation or a computer program (Bello et al., 2017; Runarsson & Jonsson, 2000; Orchard & Wang, 2016; Bengio et al., 1994). A handful of L2O methods tried to search for good symbolic rules from scratch, using evolutionary or reinforcement learning (Bello et al., 2017; Real et al., 2020; Runarsson & Jonsson, 2000; Orchard & Wang, 2016; Bengio et al., 1994). Unfortunately, those direct search methods quickly become inefficient when the search space of symbols becomes large. The open question remains as : *how to strike the balance between the tractability of finding an update rule, and the effectiveness, memory efficiency and interpretability of the found rule?*

In what follows, we will unify symbolic and numerical optimizers with a synergistic representation and analysis framework. In brief, we stick to numerical predictors at meta-training for its modeling flexibility and ease of optimization. We then convert numerical predictors to symbolic rules, which preserve almost the same effectiveness yet have negligible memory overhead and clear explainability. Facilitated by the new symbolic representations, we will develop a new suite of L2O interpretability metrics, and can meanwhile scale up L2O to much larger optimization tasks.

### 3.3 A SYMBOLIC DISTILLATION FRAMEWORK FOR L2O

We will first demonstrate how to convert the update predictor $g$: $\mathbf{x}_{t+1} = \mathbf{x}_t - g(\mathbf{z}_t, \phi)$ into *symbolic* forms that preserve the same effectiveness. Contrasting with numerical forms, symbolic rules bear two advantages: (i) being **white-box** functions and enabling **interpretability** (see Section 3.3); and (ii) being much **lighter**, e.g., unlike RNNs that have to carry their own weights and hidden states, which removes the memory bottleneck for **scaling up** to large optimization tasks (see Section 3.4).

To achieve this goal, we leverage a classical tool of symbolic regression (SR) (Cranmer, 2020), a type of regression analysis that searches the space of mathematical expressions to find an equation that best fits a dataset, both in terms of accuracy and simplicity. Different from conventional regression techniques that optimize the parameters for a pre-specified model structure, SR infers both model structures and parameters from data. Popular algorithms rely on genetic programming to evolve from scratch (Runarsson & Jonsson, 2000; Gustafson et al., 2005; Orchard & Wang, 2016). However, SR algorithms are slow on large problems and rely on many heuristics, if evolving from scratch (Runarsson & Jonsson, 2000; Gustafson et al., 2005; Orchard & Wang, 2016). They will become highly inefficient if further being entangled with the rule search, like existing attempts demonstrated (Real et al., 2020; Bello et al., 2017; Runarsson & Jonsson, 2000; Orchard & Wang, 2016).

Inspired by the idea of "knowledge distillation" (Gou et al., 2021), we propose a *symbolic distillation* approach that **decouples** the rule search step (in the numeric domain) and the subsequent representation step (to the symbolic domain). We will first learn a numerical predictor $g$ using existing L2O methods (e.g., RNN-based), and then apply SR to fit a symbolic form that approximates the input-output relationship captured by $g$. In this way, the former step can stay with the more efficient gradient-based search in a continuously parameterized space.

Specifically, we will generate an off-line database $\mathcal{D}$ of $(input, out)$ pairs, then to train SR to minimize a fitting loss on $\mathcal{D}$. Each pair is obtained from running the learned $g$ on an optimization task: $input$ is a sequence of $\mathbf{z}_t$ generated during optimization within the fixed unrolling length $T$; and $output$ is the corresponding sequence of the RNN teacher $g(\mathbf{z}_t, \phi)$. The training procedure of SR is to iteratively mutate a population of equation candidates, and filter out the better performing ones to come up with better equations. The final output of SR is a population of best equations composed of variables from $input$ that best fit $output$ of its RNN teacher $g(\mathbf{z}_t, \phi)$ on the target task distribution, using operaters coming from the pre-defined search space.

### 3.4 Concretizing the Symbolic Distillation Framework

To make symbolic distillation for L2O executable, two open questions have to be defined: **(i) "what to fit"**, i.e., what symbolic operators we use as building blocks to compose the final equation; and **(ii) "fit on what"**, i.e., what information we consider as SR input operands. These questions are open-ended and problem-dependent, and can be rather inconsistent among different L2O methods. To make our approach more generalizable, we choose relatively straightforward options for both.

**For the first question**, the selection of symbolic operators to use, we adopt the following set: $\{+, -, \times, /, (\cdot)^2, \sqrt{\cdot}, \exp(\cdot), x^y, \tanh, \operatorname{arcsinh}, \sinh, \operatorname{relu}, \operatorname{erfc}\}$. The first eight operators are adopted since they are the basic building blocks of several traditional optimizers such as Adam/-Momentum, and the rests are adopted since they are either the nonlinear activation functions used by typical L2O numerical predictors, or convey certain (update) thresholding effects which often expected and empirically used in existing deep learning optimization (Sun, 2019). For a few operators whose input ranges are constrained to non-negative only, we extend them to be valid for any real-valued input (e.g., $\sqrt{x}$ is extended to $\operatorname{sign}(x) * \sqrt{|x|}$, and $x^y$ is extended to $\operatorname{sign}(x) * |x|^y$, etc).

**For the second question**, we use the exact input variables of the numerical L2O (e.g., coordinate wise gradients) as a default option throughout this paper. To allow for further flexibility, we also enable to expand the input operand set, to provide additional optimization-related features and let the SR procedure decide which to use or not. On a high level, we model the obtained symbolic equation as a Finite Impulse Response (FIR) filter, and set a maximum horizon $T$ of the input variables. In the following parts, unless otherwise specified, $T$ is set to 20 (e.g., the latest 20 steps of input variables are visible to the rule). To facilitate capturing inter-variable relationships, we re-scale all input variables so that all of them have the uniform scale variances of 1.

We conduct a proof-of-concept experiment to distill a learned rule using the LSTM-based L2O method (Andrychowicz et al., 2016). The input sequence is $\mathbf{z}_t = \{\nabla f(\mathbf{x}_t,) \nabla f(\mathbf{x}_{t-1}), ..., \nabla f(\mathbf{x}_{t-T+1})\}$, i.e., the current and past gradients within an unrolling length $T = 20$. The L2O meta-training was performed on multiple randomly-initialized LeNets on the MNIST dataset. To fit SR, we adopt a classical genetic programming-based SR approach (Koza, 1994). The SR algorithm can score the equation complexity level as a function of the operator number, operator type diversity, and input variable number used (Cranmer et al., 2020). Hence, we can apply a range of complexity levels, and obtain a group of SR fitting rules from simple to complicated. One example of our distilled symbolic rules is displayed as ($c_i$, $i = 0,..., 19$, are scalars, omitted for simplicity):

$$\Delta x_t = \underbrace{0.013 e^{-0.835 \operatorname{asinh}(t)}}_{\text{Time-decaying step size}} \left( \operatorname{erfc} \left( \sinh\left(\nabla f(\mathbf{x}_t)\right) + \underbrace{\sum_{i=0}^{19} c_i \tanh\left(\nabla f(\mathbf{x}_{t-i})\right)}_{\text{"momentum"}} \right) - 1 \right) \qquad (2)$$

### 3.5 Interpretability from Symbolic L2O Representations

For general ML models, the interpretability is referred to as *the ability of the model to explain or to present in understandable terms to a human* (Doshi-Velez & Kim, 2017). When it specifically comes to an L2O model, its interpretability will be uniquely linked to the immense wealth of domain knowledge in optimization, from rigorous guarantees to empirical observations (Nocedal & Wright, 2006). We hence define **L2O interpretability** as *how well a learned optimizer's behavior can align with the optimization domain knowledge, or be understandable to the optimization practitioners.*

In fact, one may easily notice that the symbolic forms of L2O possess certain levels of "baked-in" interpretability. For example, the example update rule (2) immediately supplies a few interesting observations: L2O discovers a momentum-like historic weight averaging mechanism, a time-decaying step size, as well as several normalization-like or (nonlinear) gradient clipping operations such as $\mathrm{erfc}, \mathrm{sinh}$ and $\mathrm{tanh}$. All these are human-understandable as they represent well-accepted practices in deep learning optimization (Sun, 2019).

Our target is to establish a more principled and generally applicable L2O interpretability framework. The new framework will be dedicated to capturing L2O characteristics, quantifying their behaviors, and meaningfully comparing different learned optimizers. We propose two novel metrics that dissect L2O along two dimensions: (i) *Temporal Perception Field* (TPF), i.e., how long an input sequence $\mathbf{z}_t$ (past optimization trajectory) the learned optimizer "effectively" takes advantage of; and (ii) *Mapping Complexity* (MC), i.e., how complicated a predictor model $g$ needs to "effectively" be.

The estimations of TPF and MC are both facilitated in the symbolic domain. For TPF, take Eqn. (2) for example again: the predicted update at time $t$ involves a weighted (nonlinear) summation of past gradients $\nabla f(\mathbf{x}_{t-i}), i = 0, \cdots, T$ (set as 20). $c_i$ can be seen as importance indicators for $\nabla f(\mathbf{x}_{t-i})$, and a preliminary idea to define TPF is simply the weighted mean of historical lengths:

$$\text{Temporal Perception Field} = \sum_{i=0}^{T-1} \frac{|c_i|}{\sum_{i=0}^{T-1} |c_i|} \cdot i \qquad (3)$$

One challenge will be to estimate TPF when more complicated symbolic forms arise, e.g., the update rule has more heterogeneous compositions such as $a_i \left(\nabla f(\mathbf{x}_{t-i})\right)^{p_i} + b_i \left(\nabla f(\mathbf{x}_{t-i})\right)^{q_i} + c_i \mathrm{tanh}\left(\nabla f(\mathbf{x}_{t-i})\right)$. In such situations, more generic designs of TPF will have to be investigated and validated such as $\sum_{i=0}^{T-1} \frac{|a_i|+|b_i|+|c_i|}{\sum_{i=0}^{T-1} |a_i|+|b_i|+|c_i|} \cdot i$: we leave as future work.

For MC, the SR approach in (Cranmer et al., 2020) has defined a complexity score for any resultant equation, that is calculated based on the total number and types of operators as well as input variables used for fitting it. Based on that tool, we set a threshold on the maximally allowable difference between the original numerical predictor and its symbolically distilled form (over some $(input, out)$ validation set), and choose the minimum-complexity symbolic equation that falls under this threshold: its complexity score is used to indicate the intrinsic mapping complexity.

The above two definitions are in no way unique nor perfect: they are intended as our pilot study effort to quantify the understanding of L2O behaviors. We will extend the above ideas to comprehensively examining the existing L2O approaches. We aim at observing the relationship between the two metrics, and their empirical connections to L2O's convergence speed, stability, as well as transferrability under task distribution shifts. Extensive results will be reported in Section 4, from which we conclude a few hypotheses, including: (i) an L2O with larger TPF will converge faster and more stably on instances from the same task distribution, since it exploits more global optimization trajectory structure for this class of problems; and (ii) an L2O with smaller MC will be more transferable under distribution shifts, if meta-tuning (adaptation) is performed, since tuning a less complicated predictor model might be more data-efficient. We defer more details later in section 4.2 and Appendices A, B.

From the general taxonomy of interpretable deep learning (Li et al., 2021), our approach belongs to the category of using *surrogate models*, i.e., fitting a simple and interpretable model (symbolic equation) from the original complicated one (numerical predictor), and using this surrogate model's explanations to explain the original predictions. Our new interpretability metrics and analyses will be the first-of-its-kind for L2O. The findings will provide troubleshooting tools for existing L2O methods, and insights for designing new L2O methods.

### 3.6 SCALABLE L2O TUNING WITH SYMBOLIC REPRESENTATIONS

Symbolic L2O forms can not only be used as "frozen" update rules at testing, but also be tweaked to provide light-weight meta-tuning ability. Taking Eqn. (2) for example again, we can keep the equation form but set all $c_i$ to be trainable coefficients, and continue to fit them on new data, using differentiable meta-training or other hyperparameter optimization (HPO) methods (Feurer & Hutter, 2019). This "tunability" is useful for *adapting* a learned optimizer to a shifted task distribution, such as to *larger optimization tasks* since the optimizer overhead now is much lighter compared to the

*Figure 1:* The proposed symbolic regression workflow for L2O models: The **meta training** step enjoys the ease of optimization in the neural network function representation space, and come up with a numerical L2O model; the **symbolic regression** step distills a light weight surrogate symbolic equation from the numerical L2O model; the **meta tuning** step makes the distilled symbolic equation again amendable for re-parameterization.

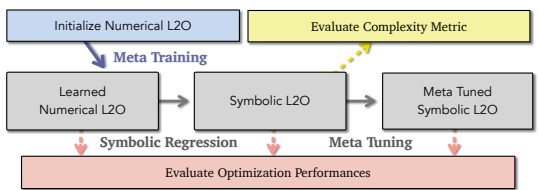

original numerical predictor. The overall procedure of *training numerical L2O → distill symbolic rule → meta tuning on new task* are illustrated in Fig. 1.

# 4 INTERPRETABILITY AND SCALABILITY EVALUATIONS FOR SYMBOLIC L2O

In this section, we systematically discuss our experimental settings and findings. Section 4.1 verifies symbolic regression's ability to recover underlying optimization rules through sanity checks. Section 4.2 leverages the interpretability of symbolic L2O to quantify several properties of the numerical L2O models. The scalability evaluation of the proposed symbolic L2O models is given in section 4.3.

## 4.1 SANITY CHECKS FOR SYMBOLIC REGRESSION ON OPTIMIZATION TASKS

As discussed in section 3.3, with the help of a few adaptative approaches (e.g., operator selection and modification, injection of inductive bias with processed features, variable magnitude re-scaling, etc.), the symbolic regression is capable to recover the underlying rules of the optimizers. To further verify the reliability of the recovery, we conduct sanity checks by regressing known optimizers, whose results are presented in table 1. The gradients are acquired from the training of ResNet-50 on Cifar-10, and the most recent 20 steps are kept for the symbolic regression inputs. The hyperparameters for the symbolic regressions are tuned to our best efforts: details could be found in Appendix C. As demonstrated in table 1, the symbolic regression faithfully reproduces the known equations.

*Table 1:* Sanity checks for SR's ability to retrieve known equations. "Seconds" corresponds to the computation time accomplished via a 2.6 GHz Intel Core i7 CPU with 16 GB 2400 MHz DDR4 Memory.

| Target Equation | Recovered Equation | $R^2$-score | Seconds |
|---|---|---|---|
| $-0.01 * g_t$ | $-0.01 * g_t$ | 1.0 | 49 |
| $\sum_{i=0}^{10}(1-0.1i)g_{t-i}$ | $g_t + g_{t-1} + 0.81g_{t-2} + 0.72g_{t-3} + 0.56 * g_{t-4} + 0.44g_{t-5} + 0.44 * g_{t-6} + 0.30 * g_{t-7} + 0.20g_{t-8}$ | 0.94 | 126 |
| $g_t^2 + g_{t-1} + 2g_{t-2} + e^{g_{t-4}}$ | $0.997g_t^2 + 2\tanh(0.47g_{t-1} + g_{t-2} - 0.101g_{t-9}) + 1.0007 + g_{t-4}$ | 0.98 | 256 |
| Momentum(0.6) | $g_t + 0.6g_{t-1} + 0.35g_{t-2} + 0.22g_{t-3} + 0.12g_{t-4} + 0.05\text{asinh}(g_{t-5} + g_{t-6})$ | 0.97 | 90 |
| Adam(0.9) | $(\sum_{i=0}^{19} a_i g_{t-i} + b_i\text{asinh}(g_{t-i}))/\sqrt{\sum_{i=0}^{19} c_i g_t^2}$ (Though some noisy asinh() appears, overall $R^2$-score remains high) | 0.89 | 879 |

## 4.2 INTERPRETABILITY OF TRADITIONAL OPTIMIZERS AND LEARNED OPTIMIZERS

In this section, we quantify the interpretability of the learned optimizers via two metrics TPF and MC which was defined in section 3.5. Without loss of generality, we consider three different problems (optimization tasks) and six representative numerical L2O backbones for the evaluation, described in Tables 2 and 3. The three problems are:

$$\mathcal{P}_1 : \min_{\boldsymbol{x}} : ||\boldsymbol{A}\boldsymbol{x} + \boldsymbol{b}||_2^2 + 0.5\boldsymbol{c}^{\mathrm{T}} \cos(\boldsymbol{x}) \qquad \mathcal{P}_2/\mathcal{P}_3 : \min_{\boldsymbol{x}} \text{CrossEntropy}\left(\text{labels}, f(\text{data}; \boldsymbol{x})\right). \quad (4)$$

$\mathcal{P}_1$ is a non-convex generalized Rastrigin function used in Cao et al. (2019), where $\boldsymbol{A} \in \mathbb{R}^{10 \times 10}$ and $\boldsymbol{x} \in \mathbb{R}^{10}$. Note that the elements in $\boldsymbol{A}, \boldsymbol{b}, \boldsymbol{c}$ are i.i.d. and sampled from $\mathcal{N}(0, 1)$ for simplicity. For $\mathcal{P}_2/\mathcal{P}_3$, we acquire the data and labels from the MNIST dataset, and $f$ denotes a MLP with Relu activation. The $\boldsymbol{x}$ indicates the parameters of the MLP and the size of the neurons for each layer is $(50, 20)$ for $\mathcal{P}_2$ and $(50, 20, 20, 12)$ for $\mathcal{P}_3$, respectively.

On the other hand, the six numerical L2O backbones considered are: DM (Andrychowicz et al., 2016), RP (Lv et al., 2017), DM+CL+IL (DM with imitation learning and curriculum learning proposed in (Chen et al., 2020a)), RP+CL+IL, RP$^{(\text{small})}$ (architecture same as RP, but with fewer coefficients: the dimension of its input projection is reduced from 20 to 6, and the number of layers of the LSTM is reduced from 2 to 1), and RP$^{(\text{small})}_{(\text{extra})}$ (architecture same as RP$^{(\text{small})}$, but with extra augmenting Adam-type gradient features into the input feature set).

**Empirical observations.** We report the values of TPF and MC of the learned L2O models in table 3, and two observations could be made. First, the larger TPF, the faster convergence speed is expected (DM+CL+IL is faster than DM, and RP+CL+IL is faster than RP). Second, with more diverse features, the learned models performs better even with lower MC (RP$^{(\text{small})}_{(\text{extra})}$ achieves the best performance among the variants of RPs and DMs, and its input features are the most diverse).

*Table 2:* The comparison of input feature set, mapping function, temporal perception field and mapping complexity across several optimizers.

| Optimizers | SGD | $mom(\beta)$ | $Adam(\beta_1, \beta_2)$ | DM | RP |
|---|---|---|---|---|---|
| Input Features Set | $g_t$ | $g_t$ | $\hat{m}_t$ | $g_t$ | $\hat{m}_t, \hat{g}_t$ |
| Mapping Function | $\psi(g_t) = -\alpha g_t$ | $\begin{array}{c} m_t = \beta m_{t-1} + (1-\beta)g_t \\ \psi(g_t) = -\alpha m_t \\ \psi(g_t) = -\alpha(1-\beta)\sum_{i=0}^{\infty} \beta^i g_{t-i} \end{array}$ | $\begin{array}{c} \psi(\hat{m}_t) = -\alpha \hat{m}_t \\ \hat{m}_t \text{ defined in Eq. 5.} \end{array}$ | $LSTM(g_t)$ | $LSTM(\hat{m}_t, \hat{g}_t)$ |
| Temporal Perception Field | 0 | $\frac{\beta}{1-\beta}$ | 0 | See Table. 3 | See Table. 3 |
| Mapping Complexity | 1 | $\approx \frac{1}{2}i_0^2$, where $\beta^{i_0} = 0.05$ | 1 (when viewing $\hat{m}_t$ as input) | See Table. 3 | See Table. 3 |

*Table 3:* Top half: the Temporal Perception Field (TPF) and Mapping Complexity (MC) values of learned L2O models. The values are averaged across three different problems $\mathcal{P}_1/\mathcal{P}_2/\mathcal{P}_3$. The tuples indicate the those models that have more than one type of inputs. Bottom half: the performance of numerical L2O and their symbolic distillation counterparts which are learned on $\mathcal{P}_2$ and evaluated on $\mathcal{P}_3$. The interpretations and the plots of optimization trajectory could be found in the Appendix B.

| Numerical L2O Models | DM | DM+CL+IL | RP | RP+CL+IL | RP$^{(\text{small})}$ | RP$^{(\text{small})}_{(\text{extra})}$ |
|---|---|---|---|---|---|---|
| Temporal Perception Field | 25.2 | 27.4 | (5.3, 2.4) | (5.7, 3.1) | (5.5, 2.2) | **(4.6, 2.0, 4.1)** |
| Mapping Complexity | 182.9 | 220.6 | 104.7 | 121.4 | 80.3 | **67.0** |
| Eva Loss by numerical L2O | $0.23 \pm 0.13$ | $0.19 \pm 0.16$ | $0.12 \pm 0.07$ | $0.11 \pm 0.04$ | $0.09 \pm 0.05$ | **$0.08 \pm 0.05$** |
| Eva Loss by distilled equation | $0.33 \pm 0.26$ | $0.33 \pm 0.19$ | $0.14 \pm 0.09$ | $0.12 \pm 0.09$ | $0.10 \pm 0.06$ | **$0.09 \pm 0.04$** |

## 4.3 SCALABILITY EVALUATIONS

In this section, we evaluate the scalability of symbolic L2O on large-scale optimizees. Due to the memory burden of LSTM-based L2O models, meta-training on large-scale problems are challenging for LSTM-based L2O models (Chen et al., 2020a; Bello et al., 2017), and most L2O models that work well with small-scale optimizees fails to scale up to large-scale optimizees, such as ResNet-50 (Chen et al., 2020a; Bello et al., 2017; Andrychowicz et al., 2016; Lv et al., 2017). Thanks to the simple form and the hidden-states-free property, symbolic L2O models are able to implement fast inference with little memory overhead during the training, which brings potentials to scale up to these larger-scale tasks. In the following, we show the feasibility that a symbolic L2O model could achieve better performance than handcrafted tuned optimizers by meta-tuning it on large-scale optimizees.

In our experiments, the best performing numerical L2O model, i.e., RP$^{(\text{small})}_{(\text{extra})}$, is selected as the teacher to perform symbolic distillation. We first meta-pre-train a RP$^{(\text{small})}_{(\text{extra})}$ model on $\mathcal{P}_1$ to get a better initialization. Then, we distill it into a symbolic equation followed by meta-fine-tuning the distilled symbolic equation on the large-scale optimizees.

**Meta-fine-tuning for distilled symbolic equations.** Recall that RP$^{(\text{small})}_{(\text{extra})}$ has three types of input features. The first two features are the same as the RP model's (Lv et al., 2017):

$$\hat{m}_t = m_t v_t^{-1/2} \quad \text{and} \quad \hat{g}_t = g_t v_t^{-1/2}, \tag{5}$$

where $g_t$ denotes the gradients, $m_t = [\beta_1 m_{t-1} + (1 - \beta_1)g_t]/(1 - \beta_1^t)$, and $v_t = [\beta_2 v_{t-1} + (1 - \beta_2)g_t^2]/(1 - \beta_2^t)$. The third type of input features $\hat{n}_t$ contains a few trainable parameters, i.e., $k_1, k_2, l_1, l_2, \alpha_1$, and $\alpha_2$. The form of $\hat{n}_t$ is similar to $\hat{m}_t$'s except that $g_t$ in $m_t$ is replaced with $k_1 g_t^{1+\alpha_1} + k_2 g_t^{1-\alpha_1}$, and $g_t^2$ in $v_t$ is replaced with $l_1 g_t^{2+\alpha_2} + l_2 g_t^{2-\alpha_2}$. As discussed in table 3, the

$\text{RP}^{(\text{small})}_{(\text{extra})}$ model has the lowest MC and TPF values, and its performance is empirically verified by the output of symbolic distillation. We summarize the the output symbolic form as follows:

$$\psi(\mathcal{G}; W) = -\sum_{g \in \mathcal{G}} \sum_{\tau=0}^{L} [W]_{g,\tau} \cdot \gamma \tanh(g_{t-\tau}/\gamma) \tag{6}$$

where $\mathcal{G} = \{\hat{m}_t, \hat{g}_t, \hat{n}_t\}$ is the set of input features, $W \in \mathbb{R}^{3 \times L}$ and $\gamma$ is the trainable coefficients. Note that the coefficient $\gamma$ is only tuned during the meta-fine-tuning stage. The accuracy of the distilled equation 6 is verified since the $R^2$ score between the distilled equation and the original $\text{RP}^{(\text{small})}_{(\text{extra})}$ model is large enough ($R^2 = 0.88$). The performance of the distilled equation is also reported in table 3. Additionally, the plotted input-output curve can be referred in Appendix B).

For the evaluation, we pick three large-scale optimizees, i.e., ResNet-50, ResNet-152 and MobileNetV2, on the Cifar-10 and Cifar-100 datasets. Due to the scale of the optimizees, it poses difficulty for the previous L2O models to execute training on a single GPU. Thanks to the simple form and the hidden-states-free property, the meta-training process for our symbolic L2O only takes less than 30GB GPU memory over ResNet-50. In addition, thanks to the initialization from pre-trained $\text{RP}^{(\text{small})}_{(\text{extra})}$, it only takes one pass for the 200-epochs' meta training before the symbolic L2O is ready.

The evaluation results are reported in Fig.2 and Table 4. For the comparison, we include the following baselines: SGD optimizers with learning rate selected from (0.1, 0.01, 0.001), with momentum selected from (0.5, 0.9, 0.99), with and without the Nesterov momentum, with and without cosine learning rate annealing, and the

*Table 4:* Comparison of the meta-tuned symbolic rule and the traditional optimizers. Evaluation results are shown for optimizing a ResNet50 on Cifar10.

| Optimizers | momentum=0.5 | | | momentum=0.9 | | |
|---|---|---|---|---|---|---|
| | lr=0.001 | lr=0.01 | lr=0.1 | lr=0.001 | lr=0.01 | lr=0.1 |
| SGD (no cosine $lr$ decay) | 94.53 | 93.70 | 94.10 | 95.20 | 94.82 | 95.31 |
| SGD (with cosine $lr$ decay) | 93.25 | 94.37 | 94.87 | 92.12 | 93.57 | 94.03 |
| Symbolic L2O | | | | **95.40** | | |

Adam optimizers with learning rate selected from (0.01, 0.001). In the legend of Fig. 2, default values are: learning rate = 0.1, momentum = 0.9, without using Nesterov momentum and without using cosine learning rate annealing. The setting of different hyperparameter selection are plotted in the legend of Fig. 2. Note that the LSTM-based L2O requires significantly higher memory cost for large-scale optimizees, thus we do not include them in this section.

As we observed in Fig.2, the proposed symbolic L2O model outperforms most of the human-engineered optimizers (SGD, Adam, Nesterov, cosine learning rate decay, etc.) in three tasks, in terms of not only convergence speed but also accuracy.

*Figure 2:* The performance comparison of the meta-fine-tuned symbolic L2O and the baseline optimizers. The symbolic L2O model achieves even better performance than the laboriously tuned traditional optimizers.

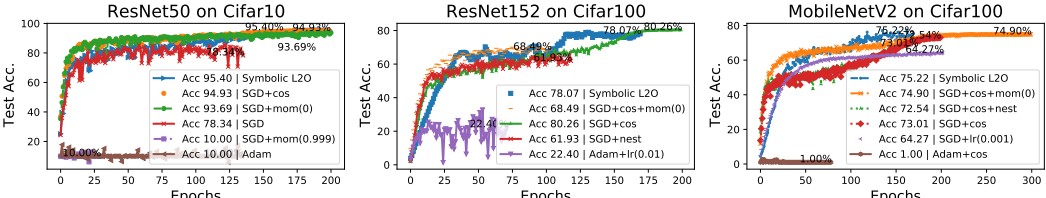

## 5 CONCLUSION

Current L2O methods face important limitations in terms of their interpretability and scalability. In this work, we attempt to overcome those limitations, by unify symbolic and numerical optimizers with a synergistic representation and analysis framework. We first numerically learn a neural network based L2O model, then distill it into a symbolic equation to link the wealth of optimization domain knowledge; the distilled equation can be meta-tuned further on testing problems. Symbolic L2O may hopefully reveal a new path to enhancing L2O to reaching the state-of-the-art practical performance. We hope our results will boost the wider adoption of L2O methods, that can eventually replace the laborious manual design or case-by-case tuning of optimizers.

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

## A  More Details Regarding The Inpterpretability

### A.1  Experimental Details

As discussed in section 4.2, we consider 6 models in the interpretability experiments: the DM, RP, and modifications of RP. The RP model differ from the DM model in two ways: RP use two gradient features (discussed in equation 5, where DM used the pure gradient as its input. Additionally, RP has an input projection layer where DM does not.

### A.2  More Detailed Discussions

**The message conveyed by the $RP^{(small)}_{(extra)}$ distillation result.** Among the comparison of table 3, the winner model is the $RP^{(small)}_{(extra)}$, which has the smallest evaluation loss. At the same time, $RP^{(small)}_{(extra)}$ is also the simplest model. This does not mean that the optimal gradient descent rule is simplest rule, but rather, the optimal gradient descent rule is the simple combination of complex gradient features, as the input feature of $RP^{(small)}_{(extra)}$ is the most complex one compared with other model, such as RP or DM.

**On the intuition behind TPF and MC: should they be large or small?** In table 3, the TPF (temporal perception field) and MC (mapping complexity) are compared together with the performance (measured in the evaluation loss) of the model. The TPF/MC are measured for certain models w.r.t. its input, hence they are blind to the concrete components of the input. We link the different behaviors of different models and their TPF and MC, with the following summaries:

- Comparing among models with the same type of input (DM vs. DM + CL + IL, or RP vs. RP + CL + IL), the larger TPF/MC lead to better evaluation loss on the testing set (newly sampled, unseen problems).

- Comparing across model with different input features (i.e., DM vs. RP, or RM + CL + IL vs. RP + CL + IL, or RP vs RP + $RP^{(small)}_{(extra)}$), the model with more complex input feature set (e.g., RP) not only outperform the model with simpler input set (e.g., DM), but they also have smaller temporal perception field and mapping complexity. Hence, improving the input feature diversity for the numerical L2O model will reduce both TPF and MC, while still leading to better performance.

**The influence of pre-training problem selection on the final performance of symbolic L2O.** Since obtaining the symbolic L2O model require a meta-pre-training step, one straightforward question may arise: what if the meta-pre-training step selected a different problem, how will the final results be affectd? To test the sensitiveness of the symbolic L2O to its meta-pre-train initialization step, we conducted ablation studies for the experiment in section 4.3 as follows.

We set up 3 different problem settings as the meta-pre-training problems, and pre-trained the $RP^{(small)}_{(extra)}$ (best performing) and the DM (worst performing) models. After that, we applied the symbolic distillation procedure, read out the distilled equation skeleton, set its coefficients as trainable, then meta-fine-tune the symbolic rule in the ResNet-50 on Cifar10. The three different problem we chose are: $\mathcal{P}_{1,2}$ introduced in section section 4.2, and $\mathcal{P}_4$, which is also minimizing the cross entropy of the MLP model, but the dataset is Cifar10 instead of MNIST.

We distill symbolic rule from the optimization trajectory data of the $\text{RP}^{\text{(small)}}_{\text{(extra)}}$ and DM models on three different problems. For $\text{RP}^{\text{(small)}}_{\text{(extra)}}$, the distilled symbolic equations take the same skeleton, which is exactly equal to equation (6), just with different coefficients. However, with the DM model, the symbolic skeletons are significantly different, as shown as follows:

$$\mathcal{P}_1 : -\text{asinh}^{0.89}\left(\log(1 + \sqrt{0.01g_t + 0.01\text{asinh}(g_{t-1}) + 0.01\text{asinh}(g_{t-2})})\right) \tag{7a}$$

$$\mathcal{P}_2 : -0.02g_t - 0.01\text{sign}(g_tg_{t-1})\sinh\left(\sqrt{\text{asinh}^{2.2}(g_t) + 0.7\text{asinh}^{2.3}(g_{t-1}) + 0.5\text{asinh}^{1.7}(g_{t-3}) + 0.2\text{asinh}^{2.1}(g_{t-4})}\right) \tag{7b}$$

$$\mathcal{P}_4 : -0.02\frac{g_t + 0.4g_{t-1} + 0.2g_{t-2} - 0.01\tanh(g_{t-3}) - 0.01\tanh(g_{t-4})}{\sqrt{g_t^2 + g_{t-1}^2 + g_{t-2}^2}} \tag{7c}$$

Note that the most desired output from the symbolic regression step is the equation skeleton, and the coefficients are subject to be further fine-tuned in the meta-fine-tune stage. Hence, the message from the ablation study results is that:

- When the meta-pre-train problem are changed, the knowledge learned by the L2O model with more diverse input features ($\text{RP}^{\text{(small)}}_{\text{(extra)}}$) could be more aligned under the same frame, compared with the L2O model with fewer input features (DM). In other words, more diverse input feature diversity of the optimizer will bring better tolerance/stability to the meta-pre-train problem selection.

- When the equation's skeleton is distilled correctly, the choice of the meta-pre-train problem does not have significant effect on the resulting symbolic L2O's performance.

**The interpretations of the distilled equations 6 and 7.** The equation 6 is the linear combination of multiple thresholded gradient features. First, these diverse features (gradient, Adam, learned exponential combination) enable higher chance to escape from flat local minima. Second, the nonlinear thresholding potentially smoothes noisy gradient signal and made the update more stable.

Equations 7 also have interpretable patterns. In $\mathcal{P}_1$, the $\text{asinh}$ is a thresholding function which clip the gradient, and inside it is $\log(x+1)$ with $x$ being the big square root. The function $y = \log(x+1)$ is close to the function $y = x$ when the magnitude of $x$ is small, where the gradient magnitude is indeed usually small. In $\mathcal{P}_2$, the equation is close to SGD in the first term $-0.02g_t$. If the sign of $g_t$ and $g_{t-1}$ is the same, i.e., two most recent gradient holds the same sign, then $\text{sign}(g_tg_{t-1}) = 1$, it descent more, otherwise it descent less. In $\mathcal{P}_4$, the equation is similar to Adam with additional $\tanh$ thresholding to certain components.

**How much adaptation is needed during meta-fine-tune.** This experiment follows the same setting of the above adaptation experiments for DM and $\text{RP}^{\text{(small)}}_{\text{(extra)}}$: meta-pre-train on $\mathcal{P}_1/\mathcal{P}_2/\mathcal{P}_4$, and meta-fine-tune and evaluate on the ResNet-50 on Cifar10. To help understand how much improvement is needed in fine-tuning, the evaluation performances (measured in test set accuracies) before and after the fine-tuning step are provided in table 5.

From the table, it can be seen that the symbolic rule from $\text{RP}^{\text{(small)}}_{\text{(extra)}}$, which have smaller MC, transfers better than DM. High complexity model DM, on the other hand, are less stable for different meta-pre-training problem.

| Optimizers | DM model | | | $\text{RP}^{\text{(small)}}_{\text{(extra)}}$ model | | |
|---|---|---|---|---|---|---|
| Meta-pre-trained from | $\mathcal{P}_1$ | $\mathcal{P}_2$ | $\mathcal{P}_4$ | $\mathcal{P}_1$ | $\mathcal{P}_2$ | $\mathcal{P}_4$ |
| Before meta-fine-tune | 13.4 | 70.0 | 83.2 | 91.3 | 93.8 | 94.4 |
| After meta-fine-tune | 10.0 | 70.7 | 84.5 | 95.4 | 95.3 | 95.5 |

*Table 5:* Comparison for before and after meta-fine-tuning. The values are test accuracies of the symbolic equation distilled from DM and $\text{RP}^{\text{(small)}}_{\text{(extra)}}$, when meta-pre-trained on $\mathcal{P}_1/\mathcal{P}_2/\mathcal{P}_4$.

**Why could the symbolic distillation provide better interpretability.** With regard to our claim in section 3.3: *white-box functions have better interpretability and lighter weight*, one question may

arise: the interpretability and light weight properties do not seem unique to symbolic optimizers, but rather, they seem unique to simple optimizers.

Indeed, the simpler model will naturally hold better interpretability than the complex model. However, whether the rule learned by the numerical optimizer is simple or not is not revealed by the numerical optimizer itself: it is only after the proposed symbolic distillation is applied can people realize that the rule learned by RP is simple (has small MC). It is also only after the symbolic distillation uncovered this fact can people safely simplify the RP model into the RP$^{\text{(small)}}$ model, to make it lighter weight and obtain better interpretability and better performance (see table 3).

In other words, the interpretability indeed belongs to simple optimizers, but a numerical optimizer can not tell the simplicity of the rules it has learned, prior to applying the proposed symbolic distillation. In this sense, the proposed symbolic distillation brings better interpretability to numerical L2O.

**The running time of the proposed method.** The proposed symbolic distillation procedure contains the following components: the meta-pre-training, SR step, meta-fine-tuning, and the final evaluation to optimize the large scale problem. The execution time for each step is provided in table 6. The proposed TPF and MC metrics are all simple calculations based on the SR results, hence they are instantly available once the SR results are out. The execution times for SR and computing TPF/MC are measured on the 2.6 GHz Intel Core i7 CPU with 16 GB 2400 MHz DDR4 Memory, and other other computation times are measured on the Nvidia A6000 GPU.

| Step | Meta-pre-train | SR | Meta-fine-tune | Final evaluation | Computing TPF/MC |
|---|---|---|---|---|---|
| Execution Time | 3 min ($\mathcal{P}_1$) - 10 min ($\mathcal{P}_3$) | 5 hours | < 20 hours | 8 hours (MobileNetV2) - 20 hours (ResNet-152) | < 1 s |

*Table 6:* The computation execution times of the experiments in section 4.3.

# B  MORE DETAILS REGARDING THE SCALABILITY

## B.1  EXPERIMENTAL DETAILS

During the symbolic regression step, the outputs are a list of distilled equation candidates. For each candidate, a unique complexity value could be measured using Cranmer (2020). During the sanity check in section 4.1, the complexity levels are chosen to be the largest output, which are relatively small (<5 for SGD, <50 for Momentum, 100 for Adam). During the experiments in section 4.3, we pick the equation with 100 complexity value, since this number gives the best fitting ability as well as the best performance (check figure 5 and table 8 for details) according to our extensive experiments.

We have also tested varying the random seeds for both meta-pre-train and the symbolic regression, and check the resulting symbolic form. Due to the low MC of the learned rule of RP$^{\text{(small)}}_{\text{(extra)}}$, the distilled equation stably shows the same form (equation 6), regardless of the random seeds.

We used 128 batch size, and in both meta-fine-tune phase and final evaluation phase, we trained the CNN optimizees for 200 epochs.

## B.2  MORE DETAILED DISCUSSIONS

**Verification of the symbolic rule fitting ability.** We verified that the distilled equation and the original optimizer are well aligned. First, figure 3 shows the fitting curve of the symbolic equation against the numerical model. In this figure, we chose a fixed time during the optimization, and plot the output of both original numerical L2O and the distilled symbolic equation. The results show that the distilled equation fits the original model well. Second, in table 7, we have distilled different optimizers, and have shown SR's fitting ability for each, indicated by $R^2$-score. The high $R^2$-score further verified the fitting ability is reliable for our tasks.

**The interpretations of the symbolic L2O ultimately used in section 4.3.** In section 4.3, we used the best numerical model (RP$^{\text{(small)}}_{\text{(extra)}}$) as the benchmark to distill the symbolic equation, and the distilled equations for RP$^{\text{(small)}}_{\text{(extra)}}$ in most cases take the form of equation (6). We note that this simple nonlinear thresholding function yielded good fitting accuracy, with the $R^2$-score 0.88. We note that the simple

*Table 7:* SR evaluations results: the fitting and optimization ability across different SR options.

| Distilled from | $mom(0.5)$ | | $Adam(0.9, 0.9)$ | | DM | | RP | | $RP^{(small)}_{(extra)}$ | |
|---|---|---|---|---|---|---|---|---|---|---|
| | $\mathcal{P}_1$ | $\mathcal{P}_2$ | $\mathcal{P}_1$ | $\mathcal{P}_2$ | $\mathcal{P}_1$ | $\mathcal{P}_2$ | $\mathcal{P}_1$ | $\mathcal{P}_2$ | $\mathcal{P}_1$ | $\mathcal{P}_2$ |
| $R^2$-score | 0.83 | 0.85 | 0.25 | 0.31 | 0.46 | 0.54 | 0.93 | 0.94 | 0.94 | 0.94 |

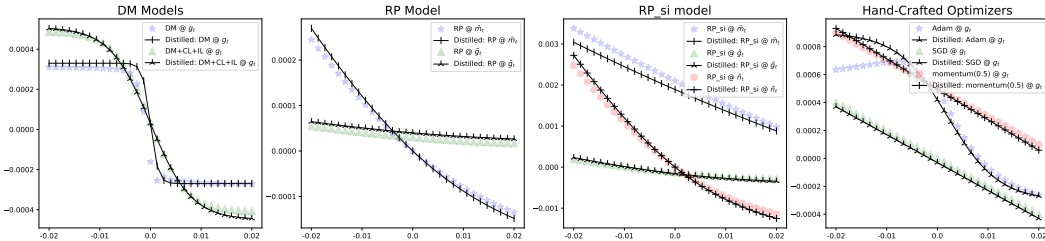

*Figure 3:* The sampled mapping functions of different optimizers w.r.t. their input feature sets. The colorful marked lines are the numerical/hand-crafted models, while the black solid lines are the distilled symbolic equations. In the line markers, "A@B" means the mapping function is A, the input of this function is B, and at the current time step, B is the only variable. It can be observed that the symbolic distillation fits the underlying optimization algorithms accurately.

form has already been verified in the small mapping complexity and temporal perception field of RL_si in table 3.

**Performance comparison of the numerical and distilled symbolic rules.** In table 3, we offered the performance comparison between the numerical rules and their distilled symbolic surrogates. To further illustrate their convergence speed, we plot the optimization trajectories for the DM model (the worst performing one) and the $RP^{(small)}_{(extra)}$ model (the best performing one) in figure 4. Specifically, the lines in figure 4 correspond to the following rows/columns in table 3: the *Evaluation Loss by numerical L2O* and the *Evaluation Loss by distilled equation*, for the *DM* and $RP^{(small)}_{(extra)}$ models.

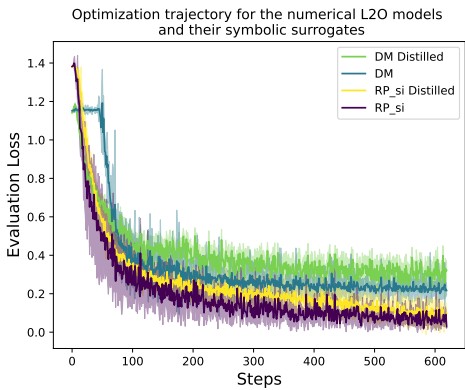

*Figure 4:* The optimization trajectories of the DM and the $RP^{(small)}_{(extra)}$ models.

**The relationship between SR complexity and fitting ability/accuracy.** The symbolic regression algorithm is able to generate a series of equations with different complexity. Intuitively, the over-simple equations under fits the L2O model, and the over-complex equations tends to overfit. The exact fitting curve is shown in Fig. 5. In this figure, the $y$ coordinate is the R2-score of the fitting performance, the higher the better, and the $x$-coordinate is the complexity of the equation. With the increase of the complexity, the fitting performance tends to first increase then decrease.

The influence of complexity to the accuracy are displayed in table 8. The experimental setting in this table is the same setting as section 4.3, and the numerical optimizer used is also $RP^{(small)}_{(extra)}$.

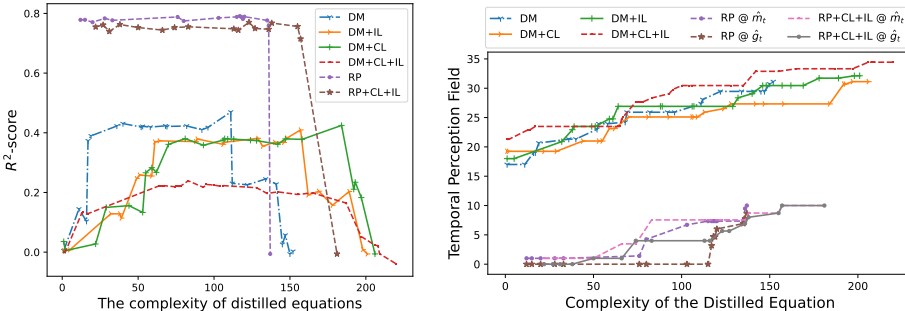

*Figure 5:* Left: the fitting performance of the equations with different level of complexity. The $R^2$-score indicates the fitting accuracy level of a single distilled equation compared with the original numerical optimizer. Right: the estimated TPF evaluated over distilled equations with different level of complexities. These figure revealed the relationship between the distilled equation complexity with the fitting ability and the estimated TPF. The values come from numerical L2Os meta-pre-trained over $\mathcal{P}_3$ (MNIST dataset with shallow MLP).

| Complexity | 10 | 100 | 200 |
|---|---|---|---|
| Accuracy | 93.5 | 95.4 | 12.0 |

*Table 8:* The evaluation accuracies of the distilled equations with different complexities from the $RP^{(small)}_{(extra)}$ model. The experimental setting is the same as section 4.3, and the values are over ResNet50 on Cifar10.

**Interpreting the hyperbolic functions presented in the distilled equations.** The hyperbolic functions appears in the distilled equations. There are two possible reasons for the occurances of the hyperbolic functions. First, the symbolic equations are learned through gradient-free random mutation, hence it is possible that such hyperbolic thresholding effect has a higher chance to reduce the uncertainty under noisy gradient data, which makes these operators to be more easily selected and kept among other mutated results. Note that for tanh()/asinh()/etc, these functions are approximately equal to $y = x$ function when the input magnitude is small (the case of tanh is plotted in figure 6). Therefore, only when the inputs have large magnitude will adding these functions result in significant difference in the output.

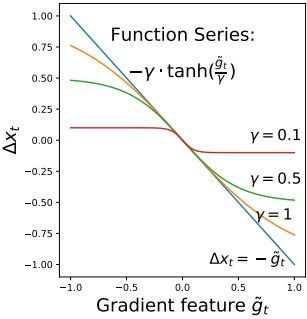

*Figure 6:* The $\tanh()$ function series. These functions are used in equation 6, where the learnable coefficient $\gamma$ determine the region of the linearity (within which the $\tanh()$ function is close to $y = x$)

Second, the tanh() functions in the distilled equations of LSTM models are straightforward to understand: the symbolic rules are the surrogate of the LSTM model, and the LSTM uses the tanh() function as its non-linear activation function. Therefore, the tanh() function in the distilled equations could be correctly reflecting the nonlinearity of the LSTM teacher's updating dynamics.

We are more interested in the distillation of a real numerical L2O model, hence we have conducted a new series of experiments, by comparing not including and including the hyperbolic functions. The distilled equations without hyperbolic functions take similar forms than with the hyperbolic functions,

| Task | Metric | AdamW Result | SymbolicL2O Result |
|------|--------|--------------|---------------------|
| SST-2 | Accuracy | 89.15 | 91.53 |
| QQP | F1/Accuracy | 86.3/89.2 | 87.8/90.8 |
| QNLI | Accuracy | 89.2 | 90.1 |
| MRPC | F1/Accuracy | 83.5/78.5 | 87.5/83.1 |
| MNLI | Matched acc. | 82.6 | 84.0 |
| CoLA | Matthews corr | 52.15 | 56.49 |

*Table 9:* GLUE fine tune performance comparisons

and are slightly different in the coefficients. The $R^2$-score does not improve by dropping these functions. This could be explained by that even given the hyperbolic functions, the equations with and without these functions are all compared and screened during the SR step, hence the resulting equation is already the best among them, though with the hyperbolic.

$$\text{Given hyperbolic (R}^2\text{ score} = 0.54): -0.02g_t - 0.01\text{sign}(g_t g_{t-1})\sinh\left(\sqrt{\text{asinh}^{2.2}(g_t) + 0.7\text{asinh}^{2.3}(g_{t-1}) + 0.5\text{asinh}^{1.7}(g_{t-3}) + 0.2\text{asinh}^{2.1}(g_{t-4})}\right) \quad (8a)$$

$$\text{Without hyperbolic (R}^2\text{ score} = 0.53): -0.02g_t - 0.01\text{sign}(g_t g_{t-1})\left(\sqrt{g_t^2 + 0.9g_{t-1}^2 + 0.8g_{t-2}^2 + 0.7g_{t-3}^2 + 0.5g_{t-4}^{2.4} + 0.4g_{t-5}^2 + 0.3g_{t-6}^2 + 0.1g_{t-7}^{2.5}}\right) \quad (8b)$$

**Large scale evaluations on GLUE tasks**

We have evaluated the symbolic rule used in section 4.3 on BERT with GLUE tasks. We compared between symbolicL2O and the AdamW. For both optimizers, we used $lr = 0.0005$. For AdamW, we used $\beta_1 = 0.9, \beta_2 = 0.999$. In all these experiments, we fine-tuned the BERT (BERT base cased model in English) for 100 epochs. For symbolicL2O, we fixed the coefficients, without meta-tuning them while fine-tuning BERT. The results are as follows. From the results, the learned symbolicL2O achieved best results across all tasks.

## C  SYMBOLIC REGRESSION PRELIMINARIES

**The psudo-code for the SR algorithm.** As described in section 1, the symbolic regression works by constantly mutating a list of equations to gradually obtain the better performing one. The core mutation algorithm of SR is provided in figure 7 as a pseudo-code.

**The hyperparameters that we chose.** There are several hyperparameters for the symbolic regression procedure that could potentially influence the performance. We list them as follows:

- The number of iterations: it means the number of executions, among which the best equations are picked. We set it to a large enough value, 300, since it shows in the experiment that any larger values leads to the similarly good results.

- The population number: it means the number of symbolic equation candidates used at each iteration. It is empirically observed that simpler models is enough to be accurately distilled with smaller population number. After tuning this parameter across all L2O models, we used the largest best population number, 200, for all models.

- Dataset size: it means the number of samples in the SR. We used 5000 samples across the experiments, which is verified to be sufficient for our purpose.

```
Algotithm: mutation operations in Symbolic Regression
function Random_Mutate (dataset):
    # cweights is the weights used to randomize mutation type.
    # If equation too big, don't add new operators
    if n >= maxsize || depth >= maxdepth
        cweights[3] = 0.0
        cweights[4] = 0.0
    end
    successful_mutation = false
    mutationChoice = rand() # randomly select one mutation type.
    # ------ Mutations ------
    while (!successful_mutation) && attempts < max_attempts
        tree = copyNode(prev)
        successful_mutation = true
        if mutationChoice < cweights[1]
            tree = mutateConstant(tree, temperature, options)
        elseif mutationChoice < cweights[2]
            # Can always mutate to the same operator
            tree = mutateOperator(tree, options)
        elseif mutationChoice < cweights[3]
            # Can potentially have a situation without success
            tree = appendRandomOp(tree, options, nfeatures)
        elseif mutationChoice < cweights[4]
            tree = insertRandomOp(tree, options, nfeatures)
        elseif mutationChoice < cweights[5]
            tree = deleteRandomOp(tree, options, nfeatures)
        elseif mutationChoice < cweights[6]
            tree = simplifyTree(tree, options) # Sometimes we simplify tree
            tree = combineOperators(tree, options)
            return PopMember(tree, beforeLoss, parent=parent_ref)
        elseif mutationChoice < cweights[7]
            tree = genRandomTree(5, options, nfeatures)
        else # no mutation applied
            return PopMember(tree, beforeLoss, parent=parent_ref)
        end
        attempts += 1
    end
    if !successful_mutation
        return PopMember(copyNode(prev), beforeLoss, parent=parent_ref)
    end
    if probChange < rand()
        return PopMember(copyNode(prev), beforeLoss, parent=parent_ref)
    else
        return PopMember(tree, afterLoss, parent=parent_ref)
    end
```

*Figure 7:* The pseudo-code for the core mutation operation used in the SR algorithm.

