# OpenReview forum: "Symbolic Learning to Optimize: Towards Interpretability and Scalability"
_ICLR.cc/2022/Conference — ICLR 2022 Poster_

### Official Review · Reviewer_VHkg · 2021-10-17

**Correctness:** 3
**Technical Novelty And Significance:** 3
**Empirical Novelty And Significance:** 3
**Recommendation:** 6
**Confidence:** 3

**Main Review:**

strengths:
1. Learning a symbolic L2O model from a numerical L2O model looks novel to me.
2. The paper shows appealing results in the comparison of other rule-based optimizers.

weakness:
1. It seems like we need to manually define a "suitable" function class for the symbolic regression.
2. I think the only reason people would first train a numerical L2O model and then distill it into a symbolic version is to apply this rule to other datasets. However, the paper lacks an experiment of the generalization of the learned symbolic L2O update rule. Moreover, I also expect a comparison (I don't see it anywhere) between the performance of numerical L2O and the distilled one.

**Summary Of The Paper:**

This paper proposed to learn a symbolic interpretable L2O model from the common numerical L2O methods. The key idea is to select the proper function class for the symbolic updating function and apply the symbolic regression to distal a light, interpretable, symbolic L2O model. In their experimental results, they first demonstrate the capability of symbolic regression for some know functions and then
they show the good distilling performance of their proposed symbolic regression method on empirical evaluation, where the learned symbolic L2O method is better than existing baseline optimization methods.



**Summary Of The Review:**

I lean to accept this paper since it is nice and appealing to distill the complicated numerical L2O model into a symbolic optimizer.
However, I think the paper would be improved dramatically by showing the generalization performance of the symbolic L2O model as well as the performance difference between the numerical L2O model and the distilled symbolic one.

---

> ### Author Response · Authors · 2021-11-18
> **Authors' response to reviewer VHkg**
>
> Dear reviewer VHkg:
>
> We appreciate your acknowledgement of this work. You mentioned in the review summary that this work is nice and appealing, yet the main weaknesses are that it lacks generalization performance as well as the comparison between the numerical L2O model and the distilled symbolic one. According to your suggestions, we have organized both the generalization and the performance comparison in the *new appendix*, together with samples of the explicit distilled expressions, the optimization trajectories, and the performance comparison before and after adaptation. We hope our updated results are found to be helpful in further understanding the performance and acknowledge the value of this paper.
>
>
> ***Q1:*** It seems like we need to manually define a "suitable" function class for the symbolic regression.
>
> ***Reply:***
> This question is a good catch. Symbolic regression needs to define the operator space to work, just as the numerical models (LSTM) needs to define an appropriate network structure. We have tested multiple different combinations of operator spaces, and have verified that the reported settings are effective to recover the underlying numerical rule.
>
>
>
> ***Q2:*** I think the only reason people would first train a numerical L2O model and then distill it into a symbolic version is to apply this rule to other datasets. However, the paper lacks an experiment of the generalization of the learned symbolic L2O update rule.
>
> ***Reply:***
> We appreciated this suggestion. We indeed have generalization experiments results, and we have organized them in table 3 in section 4.2, and table 5 in the *new section A.2*. In table 3, the below half showed the evaluation loss performance when meta-pre-trained on P2 and applied to P3. In table 5, the case of meta-pre-train on P1 and fine-tune on ResNet-50 on Cifar10 is applying the learned rule on other unseen datasets. Please kindly check them out.
>
>
>
> ***Q3:*** I also expect a comparison between the performance of numerical L2O and the distilled one.
>
> ***Reply:***
> In table 3, we have newly updated the bottom half table, where we have executed each rule, and compared between the original learned numerical rules and their distilled symbolic surrogates. Those rules are learned over P2 and evaluated over P3. We further released the optimization trajectories for 4 data points in that table, and plotted them in figure 4 of the *new appendix *B.2*. The explicit expressions for the distilled symbolic equations of the DM model and RP_si models are also provided in the *new appendix A.2*. Please kindly check them out. In summary, the distilled symbolic equation preserves almost the same performance as the numerical model, especially for the RP_si L2O model.

---

> > ### Author Response · Authors · 2021-11-22
> > **Sincerely expecting further discussions from reviewer VHkg**
> >
> >
> > Dear Reviewer VHkg:
> >
> > We want to thank you here, again, for the constructive comments and acknowledgment of this paper. We have responded to all of your questions above, and have also included more details and new experimental results in the *new appendix A and B* of the updated PDF. Could you please kindly check them out, and see if all of your concerns are properly addressed?
> >
> >
> > We would be more than happy to provide more information or clarification, should it be necessary. We hope the discussions could lead to a positive and fair assessment, to help bring new perspective in the optimization community to understand the learned rules.
> >
> > Best,
> >
> > Authors of paper 944

---

### Official Review · Reviewer_mSQF · 2021-11-02

**Correctness:** 3
**Technical Novelty And Significance:** 3
**Empirical Novelty And Significance:** 2
**Recommendation:** 6
**Confidence:** 4

**Main Review:**

## Section 2.2

I think the paper should really discuss some of the other symbolic optimizer papers in this section. A number of these are listed in section 3.2, but should really be discussed here as prior work.

Also, this [paper](https://arxiv.org/abs/2011.02159) on interpreting learned optimizers seems worth discussing.

## Section 3.1

(Minor point) Section 3.1 states that an optimizer can be either represented through a “symbolic” rule or by “numeric” computation. Ultimately, isn’t every optimizer (whether parameterized by a neural network or not) symbolic on paper, but numeric when instantiated by a computer? For example, an LSTM optimizer, if I understand correctly, would be characterized as “numeric” by this paper. But the LSTM is made up of symbolic equations! Moreover, “symbolic” optimizers such as gradient descent or momentum are numerical methods when implemented using finite precision arithmetic in a computer.

Section 3.1 also claims that “numerical predictors, especially RNN-based ones, limit L2O scalability through severe memory bottlenecks”. This paragraph confusingly discusses both memory limitations of meta-training an optimizer (the T copies of the model required for unrolled backprop), as well as the memory cost of running the optimizer itself (which depends on how the optimizer is parameterized, it is unclear to me whether this is a severe memory bottleneck). The difficulties of carrying out unrolled backprop over long sequences has been discussed in other papers (e.g., Metz et al 2019, [Understanding and correcting pathologies in the training of learned optimizers](http://proceedings.mlr.press/v97/metz19a.html); or Wu et al, [Understanding short-horizon bias in stochastic meta-optimization](https://arxiv.org/abs/1803.02021)), I think some discussion of this prior work is appropriate here.

This paper proposes scaling up learned optimizers by a multi-step process: training a neural-net-parameterized optimizer using unrolled backprop on small problems, simplifying that optimizer using symbolic regression, and then finally tuning the symbolic regression weights on larger scale problems. This implicitly assumes a hypothesis is true: that whatever algorithm a learned optimizer implements on small problems will be appropriate for large problems. I think this hypothesis should be clearly stated, and perhaps the authors could also comment on what _types_ of small optimization problems might be best for getting a good learned optimizer that yields symbolic regression rules that work well on larger scale problems.

## Section 3.2

Section 3.2 claims that symbolic optimizers are: (1) white-box functions that enable interpretability, and (2) are much lighter. I would push back against these claims, these points do not seem unique to symbolic optimizers, but rather, they seem unique to _simple_ optimizers. I can easily make up a complicated, heavy, uninterpretable symbolic optimizer.

## Section 3.3

We need much more detail about the experiments performed in section 3.3. Details of the symbolic regression algorithm? How is a “complexity level” chosen? How does the performance of the symbolic regression optimizer compare against the original, for a range of different complexity values? How sensitive are these results to random seeds? These questions could be much more thoroughly answered in the paper.

## Section 4.1

In section 4.1, how is the complexity level chosen to get these results? How much data do you need (how many optimization runs)? Can you provide more details about the optimization problem (ResNet on Cifar10, but what was the batch size / number of epochs)?

Table 1 does provide some interesting results. For example, it looks like you often get spurious hyperbolic trig functions (tanh, asinh, etc.). Could you comment more on why you think those occur? If the symbolic regression is _not_ given hyperbolic trig functions as symbols, do the r^2 values improve?

Perhaps the authors could also comment on the choice of forcing the symbolic regression to mimic the optimizer without access to _any_ state variables. This makes, for example, mimicking things like momentum much more difficult. With a state variable, momentum only needs access to the current gradient. But to approximate this, the symbolic version needs to approximate the exponential decay of coefficients for some number of gradient steps. Is it possible to give the symbolic regression access to some state variable, to make the resulting equations even more accurate and interpretable?

## Section 4.2

Table 2: How is the “temporal perception field” of Adam zero?

It would be nice to see a 2D scatter plot with mapping complexity on one axis (perhaps on a log scale), and the performance of the optimizer on the other (also perhaps on a log scale, depending on how you compute it).

## Section 4.3

Many more details are needed:
Why is pre-training done on P_1, as opposed to P_2 or P_3? If I understand correctly, P_2 and P_3 are more similar to the large scale problems (they also involve training neural networks).
Why is the performance of Adam at chance? Is there a bug, or is something badly tuned?
Are you picking the best baseline over the grid of hyperparameters, or plotting just one example from that grid in Figure 2? The text is a little unclear
Results would be more convincing with a more thorough baseline hyperparameter search.
What is the stopping criterion for the different optimizers? Are they not all trained for a fixed number of epochs?

What happens if you take one of these symbolic optimizers, and test them on a whole bunch of other different optimization problems? Do they generalize at all?

Finally, what are the symbolic rules learned after meta-tuning on the large scale problems?!? Are they interpretable? Seems like a big opportunity is missed by not showing any of the learned symbolic rules after Table 1. If the rules are interpretable, can you explain why they have good performance?

**Summary Of The Paper:**

This paper proposes using symbolic regression for making simpler, more interpretable learned optimizers.

They do this by first training a neural-network parameterized learned optimizer, and then fitting a set of symbolic equations to capture its behavior using symbolic regression. The paper shows that the method recovers aspects of the ground truth rule when applied to baseline optimizers. Then, the paper applies their technique to learn symbolic optimizers for large scale neural network training problems.

**Summary Of The Review:**

Overall, this paper addresses an important problem (better understanding what learned optimizers are doing) and does so by proposing  a new technique: using symbolic regression to fit interpretable equations to pre-trained learned optimizers.

I think the paper could be greatly improved by:
- Adding many more details about the experiments and methods. In particular, the "complexity" regularization of the symbolic rules seems like an important parameter. It's worth showing results as this parameter is swept across different values, as (presumably) there is a simplicity-vs-performance tradeoff that occurs.
- Show us the learned symbolic equations for the optimizers in section 4.3! The whole point is to have interpretable equations that we can stare at. Showing that these have good performance is one thing, but it would be much more interesting to tie that performance to particular parts of the symbolic equations.

[[ UPDATE AFTER REBUTTAL ]]
After reviewing the author response and updated manuscript, I have chosen to increase my score.

---

> ### Author Response · Authors · 2021-11-18
> **Authors' response to reviewer mSQF (Part 1)**
>
>
> Dear reviewer mSQF:
>
> We highly appreciate your acknowledgement of the importance of the problem we address, and your careful and detailed comments on our paper. You mentioned that this paper could be greatly improved by adding more details, equations, and more comparisons. Based on your comments, we have organized a *new appendix*, where summarized the experiments we previously had prior to our original submission. The *new appendix* provided answers to all of your questions with rich details, and we hope these summaries would be helpful towards a better understanding and fair and positive evaluation of this work.
>
> ***Q1:*** More papers should be discussed as prior work, such as “Reverse engineering learned optimizers reveal known and novel mechanisms”.
>
> ***Reply:***
> Thanks for this recommendation. We have added citation and analysis of this paper in the related works section.
>
> ***Q2:*** (Minor point) Isn’t every optimizer (whether parameterized by a neural network or not) symbolic on paper, but numeric when instantiated by a computer.
>
> ***Reply:***
>
> You made a fair point, but we also argue that symbolic and numerical rules are not identical. The symbolic rule emphasizes *exact* computation with expressions containing [variables](https://en.wikipedia.org/wiki/Variable_(mathematics)) that have no given value and are manipulated as symbols, although its concrete instantiation still is evaluated numerically by the computer. In comparison, a numerical rule operates on instantiated approximate float-point variables (e.g., tensor input and output) and is a black box from the symbolic view. While a symbolic rule can be naturally instantiated into a numerical rule, the vice versa (converting numerical back to symbolic) is much less straightforward.
>
> Their differences can be seen in the following example: for a symbolic rule: z = a.* x +  b.* y (a and b are scalars), {x,y,z} can be any vector, matrix or tensors as long as they are compatible in dimension. However, if z = NN(x,y) where NN is an already learned (instantiated) neural network, then in general, the dimensions of z, x and y are all fixed. That is why we pursue symbolic rules in this paper: besides interpretability, they are also lightweight and more transferrable.
>
>
> ***Q3:*** Some discussion of this prior work is appropriate in section 3.1 w.r.t. the memory bottleneck of the unrolled sequence.
>
> ***Reply:***
> We appreciate this recommendation. We have added the citations, and have rectified the statement in the corresponding discussions.
>
>
> ***Q4:*** There seems to be an implicit hypothesis in section 3.1: whatever algorithm a learned optimizer implements on small problems will be appropriate for large problems. It would be good to elaborate more.
>
> ***Reply:***
> We didn’t make the above hypothesis. This hypothesis was made by existing L2O works, but our aim is to break it. We instead hypothesized that the transferred performance of a learned optimization rule is mainly subject to two factors: **(1)** how good is the quality of the optimization rule learned over the small scale problems, and **(2)** how flexible is this rule when adapting to larger scale problems. We have released the corresponding results in the *new appendix A.2*. We have compared both DM and RP_si’s transferability in the paragraph **The influence of pre-training problem selection on the final performance of symbolic L2O**.
>
> As observed from table 5 in the appendix A.2, for the first factor, the symbolic rule of RP_si has better quality. For the second factor, the symbolic rule of RP_si transfers well, while some DM rules fail to transfer. These results are the support for our observations in section 4.2: with more diverse gradient features, the models perform better and transfer better. With these experiments, we enable the adaptation of L2O in practice, and address this cross-scale gap.
>
>
> ***Q5:*** Section 3.2, interpretability and lighter weight does not seem unique to symbolic optimizers, instead it belongs to the simple optimizers.
>
> ***Reply:***
> We agree that the simpler model will naturally hold better interpretability than the complex model. A symbolic optimizer is easily (if not always) lighter than a numerical one, since the latter has its own model parameters. As you also admit, one needs to "make up" an optimizer that goes heavy; most symbolic optimizers in use are elegantly simple.
>
> Furthermore, even discussing "simplicity" within numerical optimizers, it's not a straightforward notion before we introduce our metrics here. Numerical models don’t tell how simple they are. It is only after the proposed symbolic distillation is applied can people realize that the rule learned by RP is simple (has small MC), and to further make it lighter weight and obtain better interpretability and better performance (see table 3).
>
> In other words, you are right about the notion of "simplicity", but our method is the practical way to implement that simplicity.

---

> > ### Author Response · Authors · 2021-11-18
> > **Authors' response to reviewer mSQF (Part 2)**
> >
> > Dear reviewer mSQF:
> >
> > Due to length limitation, we continue our response session here.
> >
> >
> > ***Q6:*** More details should be added for experiments in section 3.3. Details of the symbolic regression algorithm? How is a “complexity level” chosen? How does the performance of the symbolic regression optimizer compare against the original, for a range of different complexity values? How sensitive are these results to random seeds?
> >
> > ***Reply:***
> > We appreciate these suggestions. We have added all your suggested contents in the *new appendix B*, please kindly have a check and see if more details are needed.
> >
> > - Details of the symbolic regression algorithm is put under the appendix C in figure 6.
> >
> > - The complexity level is chosen to be 100, as the complexity 100 empirically leads to the best fitting performance and the best generalizable equations, according to our extensive experiments. In section B.2, under paragraph **The relationship between SR complexity and fitting ability/accuracy**, please check figure 5 for the fitting curve, and table 8 for the performance.
> >
> > - We have tested varying the random seed for both meta-pre-train and the symbolic regression step, and check the resulting symbolic equation skeleton. Due to the low MC of the learned rule of RP_si, the distilled equation stably shows the same form (equation 6), regardless of the random seeds.
> >
> >
> > ***Q7:*** In section 4.1, how is the complexity level chosen to get these results? How much data do you need (how many optimization runs)? Can you provide more details about the optimization problem (ResNet on Cifar10, but what was the batch size / number of epochs)?
> >
> > ***Reply:***
> > Same as above, we have added all your suggested contents in the *new appendix B*.
> >
> > - In the sanity checks, the complexity levels are chosen to be the largest output, which are relatively small (<5 for SGD, <50 for Momentum, 100 for Adam).
> >
> > - We used 5000 samples across the experiments, which is verified to be sufficient for our purpose (discussed in the appendix C).
> >
> > - We used 128 batch size, and in both meta-fine-tune phase and final evaluation phase, we trained the CNN optimizees for 200 epochs.
> >
> >
> > ***Q8:*** It looks like you often get spurious hyperbolic triangle functions (tanh, asinh, etc.). Could you comment more on why you think those occur? If the symbolic regression is not given hyperbolic trig functions as symbols, do the r^2 values improve?
> >
> > ***Reply:**
> > This question is a good catch. There are two possible reasons for the occurances of the hyperbolic functions. First, the symbolic equations are learned through gradient-free random mutation, hence it is possible that such hyperbolic thresholding effect has a higher chance to reduce the uncertainty under noisy gradient data, which makes these operators to be more easily selected and kept among other mutated results, similar as the role of gradient clipping that is common in model deep network optimization. Note that for tanh()/asinh()/etc, these functions are approximately equal to y=x function when the input magnitude is small (the case of tanh is plotted in figure 6). Therefore, only when the inputs have large magnitude will adding these functions result in significant difference in the output.
> >
> > Second, the tanh() functions in the distilled equations of LSTM models are straightforward to understand: the symbolic rules are the surrogate of the LSTM model, and the LSTM uses the tanh() function as its non-linear activation function. Therefore, the tanh() function in the distilled equations could be correctly capturing the nonlinearity of the LSTM teacher’s updating dynamics.
> >
> > Besides, we have conducted a new series of experiments, by comparing not including and including the hyperbolic functions. Please kindly check the results out in the *new appendix*. The distilled equations without hyperbolic functions take similar forms compared to when with the hyperbolic functions, and are slightly different in the coefficients. The R$^2$-score does not improve by dropping these functions. This could be explained as  that even given the hyperbolic functions, the equations with and without these functions are all compared and screened during the SR step, hence dropping them does not bring further significant improvement.
> >
> >
> >
> > ***Q9:*** In section 4.1, is it possible to give the symbolic regression access to some state variable, to make the resulting equations even more accurate and interpretable?
> >
> > ***Reply:***
> > We agree that letting the symbolic regression have access to the state variable will make the resulting equation more interpretable, because the ground truth is indeed easy: the LSTM is close to 2 layer MLP that map the hidden states to the output. However, simply interpreting the transformation from hidden states to the output brings little practical help for numerical L2O: the transformation from current feature to the hidden state is still not solved. Given that, this work proposes to distill the equation from end to end.

---

> > > ### Author Response · Authors · 2021-11-18
> > > **Authors' response to reviewer mSQF (Part 3)**
> > >
> > > Dear reviewer mSQF:
> > >
> > > Due to length limitation, we continue our response session here.
> > >
> > >
> > > ***Q10:*** In table 2: How is the “temporal perception field” of Adam zero?
> > > ***Reply:***
> > > Thanks for raising this concern. The temporal perception field (TPF) is calculated w.r.t. the input feature set. For Adam, if its input feature is viewed as gradient, then the TPF is approximately $\frac{\beta_1}{1-\beta_1}$ (due to momentum in the numerator). If its input feature is viewed as $\hat{m}_t$, then since Adam rule is $-lr*\hat{m}_t$, the TPF is 0. Table 2 considered the second case (RP/RP_s/RP_si all considered the second case).
> > >
> > >
> > > ***Q11:*** It would be nice to see a 2D scatter plot with mapping complexity on one axis, and the performance of the optimizer on the other.
> > >
> > > ***Reply:***
> > > Thanks for this suggestion. We have had this experiment prior to the original submission, but held them out. In the updated PDF, we have organized them in the figure 5 and table 8 in the *new appendix B.2*, please kindly have a check.
> > >
> > > ***Q12:*** In section 4.3: Why is pre-training done on P1, as opposed to P2 or P3? Why is the performance of Adam at chance? Are you picking the best baseline over the grid of hyperparameters, or plotting just one example from that grid in Figure 2? Results would be more convincing with a more thorough baseline hyperparameter search. What is the stopping criterion for the different optimizers?
> > >
> > > ***Reply:***
> > > We appreciate these comments. We have added discussions for all of these comments in the *new appendix*.
> > >
> > > - We reported the pre-training to be done on P1, however according to our experiments, the RP_si L2O model showed stability for the selection of the pre-training problem, and the skeleton of the distilled equation is the same for pre-training on P1/P2/P3. We have reported the results for different pre-training problem selection in table 5 of the *new appendix A.2*, please kindly have a look.
> > >
> > > - The detailed configuration in figure 2 has been shown in the legends (default configurations are described in section 4.3): every legend has a fixed set of hyperparameters. We have all done grid search w.r.t. the momentum and the learning rate, which is shown in table 4. with a more thorough baseline hyperparameter search, the proposed symbolic L2O is still the best.
> > >
> > > - The stopping criterion in figure 2 is when 50 epochs did not improve the performance.
> > >
> > >
> > >
> > >
> > >
> > > ***Q13:*** What happens if you take one of these symbolic optimizers, and test them on a whole bunch of other different optimization problems? Do they generalize at all?
> > >
> > > ***Reply:***
> > > We have compared the symbolic optimizer learned from three different problems, and meta-fine-tuned on the new problems, both for RP_si and DM benchmarks. The results show that the simple rule from the RP model all generalizes well, while the DM model fails to generalize in some cases. For all of these cases, the explicit form of learned symbolic equations are also provided. Please kindly find these results in the *appendix A.2*.
> > >
> > >
> > >
> > > Q14: What are the symbolic rules learned after meta-tuning on large scale problems? Can you explain why they have good performance?
> > >
> > > ***Reply:***
> > > We have provided them in equation 6 in section 4.3. The equation is the linear combination of multiple thresholded gradient features. The simple form is expected, as the Mapping Complexity (MC) of RP_si optimizer is small thanks to its diverse features (gradient, Adam, learned exponential combination). This equation is interpretable, since these diverse features enable higher chance to escape from flat local minima. We noted the change of coefficients before and after meta-tuning, while the primitive (equation skeleton) are unchanged. These tuned coefficients make the tuned equation problem better on the new problem, while the diverse features make it still being expressive.

---

> > > > ### Comment · Reviewer_mSQF · 2021-11-19
> > > > **Thank you for your response**
> > > >
> > > > Hi,
> > > >
> > > > Thank you for your (very thorough) response. Some of my concerns have been addressed, so I will raise my score.
> > > >
> > > > I have one remaining question about the distilled equations presented in Section 4.3. If I understand correctly, the "chosen" distilled equation (equation 6) is basically a weighted average of three terms: the gradient, a running average of the first moment, and a running average of the second moment. Each of these terms is "clipped" through a tanh function as well.
> > > >
> > > > This is, overall, a pretty simple optimizer! In fact, it's simple enough to where I could see this kind of optimizer being proposed in the literature.
> > > >
> > > > Does this mean that all of these fancy learned optimizers (e.g. the LSTM or other neural networks) are unnecessarily complex? Your results suggest they are just learning clever weights on rather simple features. If that's the case, I would suggest putting some more discussion about this in the paper if possible.

---

> > > > > ### Author Response · Authors · 2021-11-19
> > > > > **Thank you for your acknolwdgement**
> > > > >
> > > > > Thank you for your response and acknowledgment!
> > > > >
> > > > > We agree with your point, and our experiments shows that given diverse gradient features (RP_si), the rule is learned to be a simple one. We consider such findings to precisely belong to this work's main merits: to "interpret" what "these fancy learned optimizers (e.g. the LSTM or other neural networks" have actually learned, and why they are only "unnecessarily complex" and can be simplified to a symbolic equation.
> > > > >
> > > > > We also note a recent paper [1] that explicitly reported similar discoveries in a spatial convex optimization setting.
> > > > >
> > > > > There are two paragraphs in our updated PDF: *The interpretations of the distilled equations 6 and 7* under appendix A.2 and  *The interpretations of the symbolic L2O ultimately used in section 4.3* under appendix B.2. We will add these discussions there.
> > > > >
> > > > >
> > > > > [1] Chen X, Liu J, Wang Z, et al. Hyperparameter Tuning is All You Need for LISTA[J]. Advances in Neural Information Processing Systems, 2021, 34.

---

### Official Review · Reviewer_ALsy · 2021-11-02

**Correctness:** 2
**Technical Novelty And Significance:** 2
**Empirical Novelty And Significance:** 1
**Recommendation:** 5
**Confidence:** 4

**Main Review:**

While I'm not very familiar with the L2O literature, I find the paper is difficult to read not due to my lack of domain knowledge but due to the writing and presentation. I suggest the authors re-organize the paper and address the following concerns:
 - Technical preliminaries should be put into section 3 as related work does not involve any formal definitions
 - Symbolic regression is the most critical component of the proposed method. I suggest the authors provide formal definitions of the algorithm and other details (figures, pseudo-code) to show how this algorithm fits into the framework. If it is borrowed from the previous work then this part should go to preliminaries.
 - Experimental results and technical details should be put into separate sections: they are tangled up in the current draft. For example:
  - Section 3.3, "We conduct a proof-of-concept experiment"
  - Section 3.4, "Extensive results will be reported in Section 4, from which we conclude a few hypotheses, including"

Terms and notations are used without explanation/definitions:
 - What is a Finite Impulse Response filter?
 - What is R^2 score in Table 1?

Mixed-use of technical terms:
 - Meta-pre-train vs meta-fine-tune vs meta-tuned vs meta-training

The faithfulness of SR:
 - Since SR learns a surrogate optimizer from a fixed snapshot of a numerical optimizer, it is critical to show that SR with the proposed operators is sufficient and accurate to recover the original optimizer.
 - However, this is demonstrated only with simple known equations in Table 1, instead of real numerical L2O models

I'm also concerned about the methodology:
 - While I agree with the motivation of improving the interpretability of the L2O model, the resulting symbolic equation is difficult to understand as well. For example, in Eq.2, why hyper-parameters are set to these values? What does T=20 mean vs. setting T to other values e.g. 30?
 - To learn the symbolic surrogate optimizer, one must first pre-train a numerical L2O model. If the pre-trained model is fully trained, doesn't the pre-training phase have the same computational cost as the traditional method?
 - If only consider the inference phase, the author claims the optimizer can be further meta-tuned for better performance. It is unclear how well does it compare with directly fine-tuning the pre-trained L2O model, as the latter is more flexible.


Experiments:
 - What does the number mean in Table 4? How can this table show that the proposed method is more scalable than the numerical ones?
 - The author could provide the running time of the proposed method showing the computational cost of all its components: L2O pre-training, SR, meta-tuning, TPF and MC computing.

**Summary Of The Paper:**

This paper proposes a symbolic L2O framework that aims to improve the scalability and the interpretability over the numeral L2O methods. The proposed framework uses symbolic regression to turn a snapshot of the numerical method into a surrogate symbolic optimizer. The resulting optimizer can be further fine-tuned with less computational costs and has a certain degree of interpretability.


**Summary Of The Review:**

In summary, this paper needs an overhaul of its writing and presentation. The proposed method is not well-justified and the important technical details are missing. The experiments are lacking and cannot support the claims made by the paper. That said, I recommend rejection.

---

> ### Author Response · Authors · 2021-11-18
> **Authors' response to reviewer ALsy (Part 2)**
>
> Dear reviewer ALsy:
>
> Due to length limitation, we continue our response session here.
>
>
> ***Q6:*** To learn the symbolic surrogate optimizer, one must first pre-train a numerical L2O model. If the pre-trained model is fully trained, doesn't the pre-training phase have the same computational cost as the traditional method?
>
> ***Reply:*** As has been discussed in the introduction section, the true value of L2O is not in its pre-training stage (meta-training), but rather in its usage (meta-testing): once pretrained over a small number of problems, L2O will be able to apply to a broader range of problems that leads to faster optimization, with almost zero marginal cost. Therefore, once an L2O model is successfully trained, regardless of the amount of computation resources used during the pre-training stage, the cost will be amortized by its reusability - this is the same philosophy as any meta-learning scheme.
>
>
> ***Q7:*** It is unclear how well does the symbolic L2O compare with directly fine-tuning the pre-trained L2O model, as the latter is more flexible.
>
> ***Reply:***
> As has been discussed in section 4.3 and has been pointed out by reviewer FzGU: the LSTM based methods require prohibitive GPU memory (due to the sequential unroll) at its meta-testing stage that forbids even just fine-tuning over large scale problems as the ResNet50, so a comparison is not available.
> The key merit of this work is to turn a pre-trained L2O model into a much lighter yet still tunable format, so that it can still continue to be adapted on much larger-scale problems during inference (meta testing) time This was summarized in our contributions in the introduction section, and we are glad to see all other three reviewers to capture this keypoint.
>
>
> ***Q8:*** How can table 4 show that the proposed method is more scalable than the numerical ones?
>
> ***Reply:***
> You misread Table 4. The *numerical* optimizer in this paper all refers to the LSTM based learnable method, while SGD or Adam are referred to as traditional optimizers. table 4 did not compare the distilled symbolic equation with numerical ones, but only with the traditional optimizers.
>
>
>
> ***Q9:*** The author could provide the running time of the proposed method showing the computational cost of all its components: L2O pre-training, SR, meta-tuning, TPF and MC computing.
>
> ***Reply:***
> The execution time for these components are all provided in table 7 in the *new appendix* A.2.
>
> In general, we believe our paper’s contributions are sound, as agreed by all other reviewers. Quoted from reviewer FzGU: “the overall framework and concept are sound.” Quoted from reviewer jXyx: “I like the idea of the paper, and overall the execution looks promising.” Quoted from reviewer mSQF: “this paper addresses an important problem (better understanding what learned optimizers are doing) and does so by proposing a new technique”. Quoted from reviewer VHkg: “I lean to accept this paper since it is nice and appealing to distill the complicated numerical L2O model into a symbolic optimizer”.
>
> We hope the newly released experimental details and results could eliminate your confusions.
>
> Best,
>
> Authors

---

> > ### Author Response · Authors · 2021-11-22
> > **Sincerely expecting further discussions from reviewer ALsy**
> >
> >
> > Dear Reviewer ALsy:
> >
> > We appreciate your time and the constructive comments in your review. We have addressed all of your concerns above with experimental results and more details, have you gotten a chance to read them?
> >
> > As a follow-up on our responses, we would like to kindly remind that the discussion period is ending soon. We hope to use this open response period to discuss the paper to solve the concerns and improve the quality of our paper. We would be more than happy to provide more information or clarification, should it be necessary, and hope that they could lead to a positive and fair assessment of this paper.
> >
> > Best,
> >
> > Authors of paper 944

---

> ### Author Response · Authors · 2021-11-18
> **Authors' response to reviewer ALsy (Part 1)**
>
>
> Dear reviewer ALsy:
>
> We truly appreciate your valuable time spent on reviewing our paper. You mentioned the two main weaknesses of this paper is the presentation, and that the conclusions are not well-justified due to the missing technical details. According to your comments, we have improved our paper presentation, and have organized and released many experimental details in the *new appendix*. We’ve answered all of your questions in detail as follows.
>
> ***Q1:*** Technical preliminaries should be put into section 3 as related work does not involve any formal definitions
>
> ***Reply:***
> Thanks for this recommendation, we have made the change accordingly.
>
>
> ***Q2:*** I suggest the authors provide formal definitions of the algorithm and other details (figures, pseudo-code) to show how this algorithm fits into the framework.
>
> ***Reply:***
> Thanks for this suggestion. We have provide the pseudo-code for the core algorithm of SR in the *new appendix C*, in figure 6.
>
>
> ***Q3:*** What is a Finite Impulse Response filter and what is R^2 score in Table 1?
>
> ***Reply：***
> The Finite Impulse Response (FIR) filter is a fundamental concept in digital signal processing. The Wiki link is [here](https://en.wikipedia.org/wiki/Finite_impulse_response). The R^2 score is the proportion of the variation in the dependent variable that is predictable from the independent variable(s). The Wiki link is [here](https://en.wikipedia.org/wiki/Coefficient_of_determination). We use the FIR filter here for an analogy: both FIR filter and the distilled symbolic equation take finite historical input sequence. In comparison, the Infinite Impulse Response (IIR) filter is an analogy to the LSTM. We did not explain in the original draft because we thought both are rather basic concepts - but we’re happy to include their explanation for more audience accessibility.
>
> ***Q4:*** It is critical to show that SR with the proposed operators is sufficient and accurate to recover the original optimizer, to demonstrate the faithfulness of SR.
>
> ***Reply:***
> We agree with the importance of the verification that the adopted SR operators are efficient and accurate. We have already shown this in our original submission: the R2-score reported in table 1 verifies that the operators are sufficient to recover high quality equations. Furthermore, we have newly released detailed verification for the learned optimizers in the updated PDF in multiple ways.
>
> In the *new appendix B.2*, table 8 and figure 3 show both numerically and graphically that the SR have high fitting accuracy. The below half of table 3 in section 4.2 and figure 4 in the *new appendix B.2* also show  both numerically and graphically that the SR is able to recover numerical model’s optimization behavior.
>
>
>
> ***Q5:*** In Eq.2, why hyper-parameters are set to these values? What does T=20 mean vs. setting T to other values e.g. 30?
>
> ***Reply:***
> Eq.2 is one sample distilled equation. The hyper-parameters are learned through the distillation procedure instead of set. The value T, maximum historical sequence length, is indeed one hyper-parameter of the symbolic regression. Actually, the historical sequence length that the model can capture exactly means the temporal perception field (TPF), which has been discussed in section 4.2. From table 3 in section 4.2, we can obtain that TPF of the RP_si model is 4.6, which is less than 20. Even when we set the maximum historical sequence length to 80 in our experiments, the RP_si can only still recover the same TPF. Given that 4.6<20, setting T to 30 or 20 will lead to the same results, except that T=30 takes more time to compute.

---

> > ### Author Response · Authors · 2021-11-26
> > **Sincerely expecting further discussions from reviewer ALsy**
> >
> > Dear reviewer ALsy:
> >
> > We hope to bring to your attention that the discussion phase is ending soon. Since we have addressed all of your concerns in the updated PDF with much more details, could you please have a look and let us know if there is any new comments?
> >
> > Best,
> >
> > Authors of paper 944

---

> > > ### Comment · Reviewer_ALsy · 2021-11-29
> > > **Thoughts on the feedbacks**
> > >
> > > Thank you for the detailed response. The revised draft definitely goes in the right direction. Some of my confusion on the proposed method and experiment setup are addressed. However, my concerns about the SR algorithm and the overall writing clarity are still not addressed.
> > >
> > > Given that the SR algorithm serves as the most important component of the proposed method, I'm surprised that a complete presentation of the proposed SR method is still absent in the revised draft:
> > > - It's unclear how the "generic mathematical operator space" in (Bello et al., 2017) is adopted for the symbolic operators.
> > > - The authors use the genetic algorithm for SR search, but it's unclear how the symbolic rules together with the numerical coefficients are represented and stored in the search space, as the space is discrete and highly complex.
> > > - There are still many details missing regarding the implementation of SR. The genetic algorithm relies on many heuristics and is known to be sensitive to the choice of hyper-parameters, yet none of these design details are provided: 1) what is your population size? 2) For mutation how do you set the mutation probability? Do you also use crossover? If so 3) what is your parent selection policy? How is crossover performed? If not 4) one needs to mention it and justify it properly.
> > > - Pseudo code in Figure 7 shows the mutation operation rather than the whole picture of the algorithm. The code is also hard to decipher: what do the parameters represent (e.g. tree, options, nfeatures)? What do all these functions do?
> > >
> > > While I appreciate the authors' significant efforts in adding new experiments in the appendix. I still find the experiments and analysis convoluted, as many of them are done under different settings without proper justifications, making it difficult to draw comparisons:
> > > - The authors used LeNet on MNIST for eq (2). Why not use P2 and P3?
> > > - Besides P1/P2/P3, the authors proposed another problem P4 in section A.2 for evaluating the influence of changing the problem sets. What is the motivation behind this?
> > > - According to A.2, distilled rules of RP_si "is exactly equal to eq(6)" but this observation conflicts with that in the main text, which states that "RP_si are all simple and take similar forms. We summarized this form as follows, and fix this form". This is concerning as manually fixing the form leads to an unfair comparison with other backbones.
> > >
> > > Other minor issues:
> > > - Mixed use of notations: predictor model g vs gradient at time t g_t
> > > - Mixed use of terms: meta-pre-train vs meta-fine-tune vs meta-tuned vs meta-training vs meta-testing
> > > - Typos: "datas", "the the datas and labels", …
> > >
> > > Overall, I'm raising my score to 5 but I'm still inclined to reject as the current draft lacks the necessary details and empirical results/analysis for its core component, which is the SR algorithm. And the presentation on experiment settings requires further polishing.

---

> > > > ### Author Response · Authors · 2021-11-29
> > > > **Thank you and Addressing further concerns**
> > > >
> > > > Dear reviewer ALsy:
> > > >
> > > > Thank you for your response and comments on improving the paper! W.r.t. the new concerns you raised, we have addressed them as follows.
> > > >
> > > > **Q1:** How the "generic mathematical operator space" in [1] is adopted for the symbolic operators.
> > > >
> > > > **Reply:** [1] proposed to learn operators from scratch, and thus defined the operands (model input), unary operators and binary operators. We followed their way in defining the search space, and similarly defined the set of operands/operators for our purpose. Since we distill from the LSTM model, our operands are natually defined as the same operands as the LSTM L2O model. For the operators, we used the same operators: $x, -x, log(x), sign(x), etc.$, and additionally use the $tanh(x), asinh(x)$, which achieves the same thresholding effect as their adopted $clip(x, 1e-5), etc.$
> > > >
> > > > [1] Bello I, Zoph B, Vasudevan V, et al. Neural optimizer search with reinforcement learning[C]//International Conference on Machine Learning. PMLR, 2017: 459-468.
> > > >
> > > >
> > > > **Q2:**  It's unclear how the symbolic rules together with the numerical coefficients are represented and stored in the search space.
> > > >
> > > > **Reply:** We recommend to check our originally uploaded supplementary materials, we have provided several intermediate results there. As an example, in `/l2o-supplementary/gpSR-samples/final-model-dm-mtil, neuron-1, var-g04-w.t-w.g-feature, target-a/hall_of_fame.csv.bkup`, one could find:
> > > >
> > > >  $log(pow(-0.927, abs(sinh(plus(0.047, mg)))))$
> > > >
> > > > This equation is the one sample output. If we were to fine tune this equation, we then replace the $-0.927$ and $0.047$ in the equation as trainable parameter $w_1, w_2$, and send it to the fine-tuning stage. All these $log, pow, etc.$ functions are differentiable and are well-defined in the code.
> > > >
> > > > **Q3:** What is the details of the  genetic algorithm ,
> > > >
> > > > **Reply:** We have already provided these information in the *appendix C* in the second paragraph, we kindly recommend to have another check. The population size is 200. The iteration number is 300. For the mutation probability, crossover and parent selection policy, we followed the stable implementation in https://github.com/MilesCranmer/PySR. Our choices of the hyperparameters are empirically found to be sufficient, and shown to be valid in our sanity checks in table 1 and table 3.
> > > >
> > > > **Q4:** What to the tree, options, nfeatures, and the functions mean in figure 7?
> > > >
> > > > **Reply:** Trees means the current tree node: since symbolic equation can be easily written as a tree form, whose in-order traversal is the equation, manipulation over a tree node leads to the most efficient manipulation of sub-components in a equation. options/nfeatures dectate the optional details in performing the corresponding actions, and other functions such as `genRandomTree`, `deleteRandomOp`, etc, simply perform operations indicated by their name. More details could be found here: https://github.com/MilesCranmer/SymbolicRegression.jl/blob/master/src/Mutate.jl.
> > > >
> > > > **Q5:** Why use LeNet on MNIST for eq (2) not P2/P3?
> > > >
> > > > **Reply:** In the section 3.4 where the eq(2) belongs to, we targeted at concretizing the SR framework, hence we used a simple problem yet the one that is composed of Neural networks: the LeNet on MNIST. First, this choice of optimization problem is simple enough for a proof-of-concept. Second, P2/P3 is extensively studied in table 3/5/7, hence we believe our experiments are consistent in the later sections which contain multi-dimensional comparisons and rich results.
> > > >
> > > > **Q6:** What is the motivation to use a new problem P4 in section A.2 for evaluating the influence of changing the problem sets?
> > > >
> > > > **Reply:** The motivation is to compare our algorithm in a more diverse dataset distributions. P2/P3 use the MNIST (though the optimizee architectures are different depth NNs), while P4 use the Cifar10 dataset.
> > > >
> > > > **Q7:** There is a concern as manually fixing the form leads to an unfair comparison with other backbones.
> > > >
> > > > **Reply:**
> > > > As discussed in section 4.3, we only "fix" the form during the "meta-fine-tune" stage. The reason that we fix it is consistent throughout the main text the the appendix: in appendix A.2, we stated "distilled rules of RP_si is exactly equal to eq(6)", and the main text states RP_si "take similar forms and we fix this form".
> > > >
> > > > On the other hand, we did not fix the form for other optimizers besides the RP_si. It is only after consistent observation (both in section 4.3 and appendix A.2) did we use a unified symbolic skeleton for RP_si fine-tuning.
> > > >
> > > > **Q8:** There are other minor issues.
> > > >
> > > > **Reply:**
> > > > We appreciate your carefulness! We have updated them in our local drafts.
> > > >
> > > > We sincerely hope our responses could help to address your concerns!
> > > >
> > > > Best,
> > > >
> > > > Authors of paper944

---

### Official Review · Reviewer_FzGU · 2021-11-03

**Correctness:** 2
**Technical Novelty And Significance:** 3
**Empirical Novelty And Significance:** 2
**Recommendation:** 6
**Confidence:** 4

**Main Review:**

The two main ideas in this paper are:
(1) One can train the neural optimizer on small problems and distill it into symbolic form. Then apply it to large problems with adaptation. This can help the interpretability and scalability of the learned optimizer.
(2) TPF and MC are proposed as two metrics to understand the optimizer.

The ideas and the overall framework are sound. However, the major issue of this paper is that the proposed method and the claims are not well-supported by the experiments.

Experiments in Sec 4.1:
- The sanity check is a good experimental setting to verify the correctness of symbolic regression. This set of experiments are meaningful but symbolic regression is not the main contribution of this paper. The main paper didn't even describe the concrete algorithm for symbolic regression. Apart from the distilled equations for known optimizers, we are more interested in seeing more examples of distill equations for the neural optimizers (such as the symbolic L2O in Section 4.3).

Experiments in Sec 4.2:
- The reported numbers seem to be insufficient to support the claims. The authors try to show that the metrics TPF and MC are linked to optimizers behavior such as the convergence and transferability. However, why are the metrics TPF and MC reported only without numbers measuring the model accuracy, convergence, generalization abilities are reported along with these metrics?
- To support some claims such as 'larger TPF will help converge faster' and 'models with smaller TPF and larger MC will be more transferable', one needs to measure the convergence and transferability of different optimizers and see whether such a linkage is true.

Experiments in Sec 4.3:
- What is the symbolic form of the distilled optimizer? This very important result is missed.
- Although the authors said RP_si model is the best performing one so they choose to distill this model. It will be also interesting to see what's the performance of distilling other neural optimizers and also the expressions of those symbolic optimizers. Otherwise, we only see one example.
- Table 4 says 'ResNet152 on Cifar10'. Is this a typo? Should it be 'ResNet50 on Cifar10'?
- Since the symbolic optimizer is meta-fine-tuned on ResNet-50 on Cifar10, it is better to compare what is the performance before and after the adaptation in order to get an idea of how much the adaptation is needed.
- Furthermore, it is also good to know what is the performance of training the symbolic optimizer from scratch on ResNet-50 on Cifar10. This can verify the necessity of distilling the optimizer from a neural optimizer. Especially, when the neural optimizer is trained on a different smaller problem, it is difficult to see whether there will be a performance drop if one trains the symbolic optimizer directly on large target problems.
- It seems the symbolic L2O does not perform the best on ResNet152 on Cifar100. Could the authors elaborate on this?
- Finally, as mentioned that the LSTM based methods require more memory so a comparison is not shown. I think for research interest it is beneficial to make such a comparison on smaller problems to see how large the performance gap is.

**Summary Of The Paper:**

This paper proposed symbolic learning to optimize (L2O), in which a neural optimizer will be trained and then distilled into a symbolic form via symbolic regression. This symbolic optimizer will then be applied to solve large problems, with some adaptation and fine-tuning.

This paper also proposed two metric called TPF and MC to understand the behavior of the optimizers.

**Summary Of The Review:**

The overall framework and concept are sound. This will be a good paper if the experiments can support the arguments. However, the current experiments have left out too many details and results for validating the effectiveness of the proposed method.

---

> ### Author Response · Authors · 2021-11-18
> **Authors response to reviewer FzGU (part 1)**
>
>
> Dear reviewer FzGU:
>
> We highly appreciate your detailed review of our work: these valuable comments are really good catches of the key interpretations of this work! We have responded to all the raised questions with details, and have organized them into the new appendix of the updated PDF. We hope the discussions and these newly released results could lead to a better understanding of the methodology and the theoretical value of this work.
>
> ***Q1:*** We are interested in seeing more examples of distilled equations for the neural optimizers (such as the symbolic L2O in Section 4.3).
>
> ***Reply:***
> This is a good suggestion. As mentioned above, we have organized our rich experiment results, including more distilled equation examples in the *new appendix A.2*, please kindly have a check. The newly released results include: the symbolic rules actually used in section 4.3 (distilled from RP_si model), the successful symbolic rules distilled from the DM model, with and without hyperbolic operators, and the unsuccessful symbolic rules distilled from DM due to overfitting.
>
> Equation used in 4.3 (please check updated section 4.3 for more justifications)
>
> $\psi(\mathcal{G};W)=-\sum_{g\in \mathcal{G}} \sum\_{\tau=0}^{L} [W]\_{g,\tau}\cdot \gamma\tanh(g\_{t-\tau}/\gamma)$
>
> Example equations from the DM model (please check appendix A.2 for more justifications):
>
> $- 0.02\frac{g_{t} +  0.4g_{t-1} + 0.2g_{t-2}  - 0.01\tanh(g_{t-3})  - 0.01{\rm tanh}(g_{t-4} ) }{\sqrt{g_t^2 + g^2_{t-1}+g^2_{t-2}}}$
>
> $- 0.02g_{t} -  0.01{\rm sign}(g_t g_{t-1})\sinh\left(\sqrt{ {\rm asinh}^{2.2}(g_t) + 0.7{\rm asinh}^{2.3}(g_{t-1})  + 0.5{\rm asinh}^{1.7}(g_{t-3})  + 0.2{\rm asinh}^{2.1}(g_{t-4}) }\right)$
>
>
> $ -{\rm asinh}^{0.89} \left( \log(1+\sqrt{0.01g_t+0.01{\rm asinh}(g_{t-1}) + 0.01{\rm asinh}(g_{t-2})}) \right) $
>
>
>
>
>
> ***Q2:*** The metrics TPF and MC are reported, but numbers measuring the model performance metrics should better be reported as well.
>
> ***Reply:***
> This is a good suggestion. These numbers are initially omitted  to avoid overflooding the readers, yet we agree they are of vital importance towards a thorough understanding of this paper and are included in the updated PDF.
>
> In table 3 in section 4.3, the below half showed the evaluation loss for both numerical and distilled symbolic equations. In the *new appendix B.2*, the optimization trajectory for RP_si and DM in table 3 is plotted, in order to show the convergence rate. The results showed that the symbolic surrogate is close to the numerical model, and the newly proposed RP_si is the best across the models.
>
>
>
> ***Q3:*** It will be also interesting to see what's the performance of distilling other neural optimizers and also the expressions of those symbolic optimizers.
>
> ***Reply:***
> Thanks for making this suggestion. Similar as above, the distillation results of other neural optimizer (DM) are shown, both in the performance and their explicit symbolic form in the *new appendix A.2*.
>
> ***Q4:*** Table 4 says 'ResNet152 on Cifar10'. Is this a typo? Should it be 'ResNet50 on Cifar10'?
>
> ***Reply:***
> Thanks for careful reading! Indeed it is a typo intended to be ResNet50 on Cifar10. We’ve fixed it.
>
> ***Q5:*** It is better to compare what is the performance before and after the adaptation in order to get an idea of how much the adaptation is needed.
>
> ***Reply:***
> We appreciate this suggestion. The results of comparing before and after meta-fine-tune are released in table 5 in the *new appendix A.2*. We compared two numerical L2O models: the best model PR_si and the worst model DM, each meta-pre-trained on three different problems and meta-fine-tuned on for ResNet50 on Cifar10.
>
> From table 5, it can be seen that the symbolic rule from RP\_si, which has smaller MC, transfers better than DM. High complexity model DM, on the other hand, are less stable for different meta-pre-training problems.

---

> > ### Author Response · Authors · 2021-11-22
> > **Sincerely expecting further discussions from reviewer FzGU**
> >
> >
> > Dear Reviewer FzGU:
> >
> > We would like to thank you again here, for the constructive comments in your review. We hope to use this open response period to discuss the paper to solve the concerns and improve the quality of our paper. Have you gotten a chance to read our responses below, which include several new experimental results to address all of your concerns?
> >
> >
> > We would be more than happy to provide more information or clarification, should it be necessary, and hope that they could lead to a positive and fair assessment of this paper.
> >
> >
> > Best,
> >
> > Authors of paper 944

---

> ### Author Response · Authors · 2021-11-18
> **Authors response to reviewer FzGU (part 2)**
>
> Dear reviewer FzGU:
>
> Due to length limitation, we continue our response session here.
>
>
> ***Q6:*** It is also good to know what is the performance of training the symbolic optimizer from scratch on ResNet-50 on Cifar10.
>
> ***Reply:***
> Obtaining symbolic equations is a computationally challenging problem, as it requires searching over a gigantic discrete space. On a scale as large as ResNet50 on Cifar10, it is generally not feasible to train a symbolic equation from scratch. As a comparison, another prior work [1] trained symbolic rule from scratch, hence their method can only support rather small neural nets: a 2 layer 3x3 ConvNet on Cifar10. We are also unable to directly search symbolic rules over the much larger ResNet 50/Cifar10 scale.
>
> Indeed this motivates our most important contribution in this paper: given the inefficiency of training symbolic rule from scratch, we turn to first numerically learn an optimizer, then symbolically distill it. In this way, both two steps become feasible, and the outputs are verified to fit the learned numerical rule well.
>
> [1] Bello I, Zoph B, Vasudevan V, et al. Neural optimizer search with reinforcement learning[C]//International Conference on Machine Learning. PMLR, 2017: 459-468.
>
>
> ***Q7:*** It seems the symbolic L2O does not perform the best on ResNet152 on Cifar100.
>
> ***Reply:***
> Thanks for raising this concern. We point out that (1)  the two’s performance gap was only very marginal here; and (2) even in this occasion, a traditional optimizer outperformed L2O,  the empirical value and improvement of the proposed symbolic L2O is still sound.
>
> Specifically, the specific configuration of this single performant traditional optimizer (SGD & lr=0.1 & cosine lr decay & momentum = 0.9 & without using nestrov momentum) is not an instant choice. We performed grid search for lr and momentum (shown in table 4), as well as exhaustive search for cosine lr decay, optimizer type and momentum hyperparameters, to obtain one best-performing optimizer in one single problem. It is a costly search process.
> In contrast, the symbolic L2O is only trained once to get the near optimal performance across three dramatically different problems. This comparison, in fact, endorses how robustly the symbolic L2O works with high automation.
>
>
> ***Q8:*** The LSTM based methods require more memory so a comparison is not shown. For research interest it is beneficial to compare LSTM and the distilled symbolic rule on smaller problems to see how large the performance gap is.
>
> ***Reply:***
> As mentioned above, the gap between numerical and the symbolic optimizer on smaller problem is presented in table 3 (updated)  in section 4.2, and figure 4 in the *new appendix B.2*.

---

> > ### Author Response · Authors · 2021-11-26
> > **Sincerely expecting further discussions with reviewer FzGU**
> >
> > Dear reviewer FzGU:
> >
> > We noticed that the main concern you left in main review is "the current experiments have left out details and results for validating the effectiveness of the proposed method", and in response to that, we have largely enriched more details in the *new Appendix A/B* and our responses above.
> >
> > We would like to bring to your attention that the discussion phase is ending soon, could you please check our responses and the updated PDF, and let us know your evaluation of this work?
> >
> > Looking forward to your reply!
> >
> > Best,
> >
> > Authors of paper 944

---

> > > ### Comment · Reviewer_FzGU · 2021-11-28
> > > **Thanks for the response. A question regarding the TPF and MC**
> > >
> > > Thank the authors for their very detailed response. Some of my concerns are addressed, but I still have a question regarding the TPF and MC.
> > >
> > > The paper claims that
> > >
> > > "we conclude a few hypotheses, including:
> > >
> > > (i) an L2O with larger TPF will converge faster and more stably on instances from the same task distribution, since it exploits more global optimization trajectory structure for this class of problems; and
> > >
> > > (ii) an L2O with smaller MC will be more transferable under distribution shifts, if meta-tuning (adaptation) is performed, owing to the predictor model’s larger flexibility an L2O with smaller MC will be more transferable under distribution shifts, if meta-tuning (adaptation) is performed, owing to the predictor model’s larger flexibility."
> > >
> > > The new result in Table 5 can roughly support claim (ii) regarding the relation between transferability and MC. However, regarding the claim (i) which says "an L2O with larger TPF will help converge faster", I still could not find the empirical verification. The authors mentioned the optimization trajectory in Appendix B.2. However, it does not seem to be consistent with the claim. Please correct me if there is any misunderstanding. As I see from Table 3, DM has larger TPF scores than RP_si. As suggested by TPF score, DM should converge faster and moe stably on instances from the same task distribution?

---

> > > > ### Author Response · Authors · 2021-11-29
> > > > **Addressing the further question**
> > > >
> > > > Dear reviewer FzGU:
> > > >
> > > > Thanks for the response. In table 3, the fact that support claim (i) is that DM+CL+IL performs better than DM, and RP+CL+IL performs better than RP, while the "+CL+IL" lead to larger TPF for both DM and RP.
> > > >
> > > > We note that for TPF, it is not appropriate to directly compare between DM and RP, since they have different input feature type/diversity.  The claim (i) is actually discussed under the context that two models have the same input feature types. Correspondingly, the claim (ii) is not discussed under this context, since the lower MC is exactly brought from the difference of the input feature diversity. We have locally updated this clarification in the paper PDF.
> > > >
> > > > We hope our responses address your concern!
> > > >
> > > > Best,
> > > >
> > > > Authors of paper 944

---

> > > > > ### Comment · Reviewer_FzGU · 2021-11-29
> > > > > **Thanks for the further clarification**
> > > > >
> > > > > Dear Authors,
> > > > >
> > > > > Thank you for the further clarification. I think the revised paper has improved so I raise my score to 6.

---

> > > > > > ### Author Response · Authors · 2021-11-30
> > > > > > **Thank you**
> > > > > >
> > > > > > Dear reviewer FzGU:
> > > > > >
> > > > > > We are so glad that our response was able to turn your assessment into a more positive one. Thanks so much!
> > > > > >
> > > > > > Regards,
> > > > > >
> > > > > > Authors of paper 944

---

### Official Review · Reviewer_jXyx · 2021-11-06

**Correctness:** 3
**Technical Novelty And Significance:** 3
**Empirical Novelty And Significance:** 3
**Recommendation:** 6
**Confidence:** 4

**Main Review:**

# Stengths

* The paper tackles and important problem with a sound approach, which is presented well overall.
* The paper improves tuned optimizers on  ResNet50 + CIFAR10 and MobileNetV2 + CIFAR100.
* It is great to see that an optimizer on a toy task can transfer well to CIFAR10/100.

# Weaknesses

* What is the distilled symbolic rule (before tuning the constants) that you use in Section 4.3? Can you provide its explicit form (perhaps masking some constants would make sense for clarity purposes).
* I'd expect more interpretability analysis about the symoblic rule used in Section 4.3.
*  In my opinion, the paper needs experiments on other datasets to make a convincing case for scaling the method in the paper. One suggestion is to try to finetune a BERT-like architecture on GLUE, starting from the distilled symbolic rule used in Section 4.3.
* What's the intuition behind TPF and MC? Do we want them to be large or small? From Table 3, it seems like you want them to be small for better transferability; that makes sense to me.
* How important is the toy task to the results in section 4.3? Was it crucial that you used $\mathcal{P}_{2,3}$ to select the best symbolic distillation? I'd expect some ablations here.

## Minor.

* page 3: You should use \citet in "(Chen et al., 2017) takes the optimizees'..."
* Figure 1: make the green arrows longer and reposition the "Symbolic Regression" and "Meta Tuning" texts, so that it's clear to what arrow the texts correspond ot.
* Broken equation link in Table 2.
* page 8: "better performance better performance" <- "better performance"
* Table 3: can you report standard deviations as well?
* No evidence in the main text that "RP models generally have better performance". Can you provide detailed results to convince the reader?
* page 9: learn <- learning; distill <- distilling
* It seems like the authors were not able to complete the appendix. Completing the appendix is an important task, because for this empirical paper hyperparameters and other details about the L2O methods, and ablation studies, would be very useful.

**Summary Of The Paper:**

This is an ambitious paper that tackles an important problem: how to train general-purpose and data-driven optimization frameworks. The main idea of the paper is to explore symbolic learning to optimize (L2O) by distilling a numerical L2O optimization rule into a symbolic rule. The motivation for using a symbolic distillation is to provide interpretability and scalability of the trained optimizers. The contributions of the paper are as follows: introduction of the idea of symbolic distillation in the L2O literature; tools to interpret trained optimizers; results on larger-scale optimization tasks.

**Summary Of The Review:**

My recommendation is 5: marginally below the acceptance threshold. I like the idea of the paper, and overall the execution looks promising. However, I think that some important details and experiments are needed before the paper is published. If the authors address most of my questions/ comments, and if no obvious red flags appear, I would be very happy to update my recommendation.

--post rebuttal

Thank you for the clarifications and the new experiments. I will raise my score to 6. However, I recommend better tuning of AdamW for the GLUE experiments, because the baselines are rather weak, i.e. not all tasks match the results in Table 1 of the BERT paper (https://arxiv.org/pdf/1810.04805.pdf). I would also suggest that the authors add the new experiments in the appendix and fix the typos in the blue text, e.g. missing space before citation and missing closing parenthesis. Also, in the appendix when you discuss the interpretability of the equation from P_1, I think you mean "asihn" instead of "sihn".

Finally, I would like to note that the paper seems acceptable only after extensive experimental details were made available in the Appendix and the blue text. It's unfortunate these details were missing originally, but I am glad they are available now :).

---

> ### Author Response · Authors · 2021-11-18
> **Authors' response to reviewer jXyx**
>
> Dear reviewer jXyx:
>
> We appreciate your acknowledgement on the importance of this research problem. In response to your concerns, we have improved our paper by adding more details, and have explained them as follows.
>
> ***Q1:*** The exact form of symbolic rule used in Section 4.3, and more interpretability analysis is expected.
>
> ***Reply:***
> Thanks for this suggestion. We omitted these equations in the original version to avoid over-flooding the reader given the displayed equation example (2). We agreed that this piece is important, and have updated it in section 4.3 with boldface and below.
>
> $\psi(\mathcal{G};W)=-\sum_{g\in \mathcal{G}} \sum\_{\tau=0}^{L} [W]\_{g,\tau}\cdot \gamma\tanh(g\_{t-\tau}/\gamma)$
>
> where $\mathcal{G}$ is the gradient feature set, which contains 3 types of features for RP_si. The reason that leads to the efficiency of the equation is thoroughly analyzed in *appendix A.2*.
>
> In short, we used the best numerical model (RP_si) as the benchmark to distill the symbolic equation, and the distilled equations for RP_si in most cases take the form of equation (6). We note that this simple nonlinear thresholding function yielded good fitting accuracy (R2-score is 0.88), which has already been verified in the small mapping complexity and temporal perception field of RL_si in table 3.
>
>
> ***Q2:*** The paper needs experiments on other datasets to make a convincing case for scaling the method in the paper.
>
> ***Reply:***
> We agree with and appreciate this suggestion. We have added experiments on transferring from $\mathcal{P}_1$, $\mathcal{P}_2$, $\mathcal{P}_4$ to the large scale problem ResNet-50, all of which have different problems for training and evaluation, which demonstrates the scalability of our methods, please kindly check the appendix A.2 for details. Also according to your recommendation, we have started working on testing with large scale problems as BERT (also trying GPT2). Due to this short time frame and the bandwidth of our developers, we were not able to guarantee offering thorough experimental results. We hope to report new results whenever the successful development is achieved.
>
> ***Q3:*** What's the intuition behind TPF and MC, do we want them to be large or small from Table 3?
>
> ***Reply:***
>
> This question is a good catch. Note that TPF (temporal perception field) and MC (mapping complexity) are measured for certain models w.r.t. its input, hence they are blind to the concrete components of the input. Hence the full messages are summarized as:
>
> 1. If we compare models with the same type of input (DM vs. DM + CL + IL, or RP vs. RP + CL + IL), then the larger TPF/MC will lead to better performance.
>
> 2. On the other hand, according to table 3, the best model is RP_si, which has the smallest TFP and MC, but takes the most complicated input feature (its inputs are detailed in section 4.3 before equation (6)). Improving the input feature diversity for the numerical L2O model will reduce both TPF and MC, while still leading to better performance. These discussions are organized in the new appendix.
>
>
> ***Q4:*** How important is the toy task to the results in section 4.3, was it crucial that you used $\mathcal{P}_{2,3}$ to select the best symbolic distillation?
>
> ***Reply:***
>
> Thanks for raising this concern. According to your suggestion, the experiments regarding the meta-pre-train problem selection are shown as follows (and in the new appendix of the updated PDF). We set up 3 different problem settings as the meta-pre-training problems, and meta-pre-trained the two numerical L2O models: the RP_si (best performing) and the DM (worst performing). After that, we applied the symbolic distillation procedure, read out the distilled equation skeleton, set its coefficients as trainable, then meta-fine-tune the symbolic rule in the ResNet-50 on Cifar10. The three different problem we chose are: $\mathcal{P}_{1,2}$ in the original paper, and $\mathcal{P}_4$, which is also minimizing the cross entropy of the MLP model, but the dataset is Cifar10 instead of MNIST.
>
> The RP_si showed simple yet stable and effective optimization rules, which have the same skeleton and only differ in coefficients, regardless of the meta-pre-train problem. On the other hand, the DM model is distilled into different skeletons for different meta-pre-train problems. Comparing these symbolic equations, RP_si’s  rule is the best performing and the best transferable one. Given the difference between DM and RP_si is that the latter has more diverse input features set, it could be observed that, more diverse input feature diversity of the optimizer will bring better tolerance/stability to the meta-pre-train problem selection. More details are to be found in appendix A.2, under the paragraph **The influence of pre-training problem selection on the final performance of symbolic L2O**.
>
> We hope our answers help to address your concerns.
>
> Authors of paper 944

---

> > ### Author Response · Authors · 2021-11-22
> > **Sincerely expecting further discussions from reviewer jXyx**
> >
> >
> > Dear Reviewer jXyx:
> >
> > We want to thank for the constructive comments in your review. As a follow-up on our responses, we would like to kindly remind that the discussion period is ending soon. We hope to use this open response period to discuss the paper to solve the concerns and improve the quality of our paper. Have you gotten a chance to read our responses below, which attempt to address all of your concerns?
> >
> >
> > We would be more than happy to provide more information or clarification, should it be necessary, and hope that they could lead to a positive and fair assessment of this paper.
> >
> >
> > Best,
> >
> > Authors of paper 944

---

> > > ### Comment · Reviewer_jXyx · 2021-11-22
> > > **Thank you for the thorough response. Some suggestions below.**
> > >
> > > At first sight, it seems that the additions are substantial, so it would take me a while to read them thoroughly. I would suggest that you mark the changes in blue to aid this process.
> > >
> > > My understanding is that this paper is written as an empirical one and that you are proposing a method to improve the work of practitioners. Thus, the question of generalization (also raised by Reviewer VHkg in Weakness 2) is an important one. Have you had a chance to fine-tune BERT on a small GLUE task, for example? You mentioned that you are working on it and I understand you may not have time to complete the experiments until the revision submission deadline. However, I think it is an important experiment given the empirical nature of the study. As Reviewer mSQF noted, the distilled equation (thank you for providing it in Section 4.3!) is simple enough to be proposed in the literature, so it would be important to show that the results are not limited to small datasets and models. I would suggest that the authors share some of their initial results on generalization during the remaining discussion period.

---

> > > > ### Author Response · Authors · 2021-11-22
> > > > **Thanks for your feedback**
> > > >
> > > > Dear reviewer jXyx:
> > > >
> > > > We appreciate your feedback! We have marked the changes in the main text as blue color. The entire appendix is newly added so it is not marked, but we also encourage to have a read.
> > > >
> > > > We are actively working one tuning BERT on GLUE with the learned optimizer. We hope to report the results soon.
> > > >
> > > > Best,
> > > >
> > > > Authors of paper 944.

---

> > > > > ### Author Response · Authors · 2021-11-26
> > > > > **BERT GLUE results are out**
> > > > >
> > > > > Dear reviewer jXyx:
> > > > >
> > > > > We have evaluated the symbolic rule used in section 4.3 on BERT with GLUE tasks. We compared between symbolicL2O and the AdamW. For both optimizers, we used $lr=0.0005$. For AdamW, we used $\beta_1=0.9, \beta_2=0.999$. In all these experiments, we fine-tuned the BERT (BERT base cased model in English) for 100 epochs. For symbolicL2O, we fixed the coefficients, without meta-tuning them while fine-tuning BERT. The results are as follows. From the results, the learned symbolicL2O achieved best results across all tasks.
> > > > >
> > > > > Please let us know if the new results helps to resolve your concerns on the scalability of the proposed method.
> > > > >
> > > > >
> > > > >
> > > > > | Task  | Metric                       | AdamW Result | SymbolicL2O Result |
> > > > > |-------|------------------------------|-------------|---------------|
> > > > > | SST-2 | Accuracy                     | 89.15       |  91.53     |
> > > > > | QQP   | F1/Accuracy                  | 86.3/89.2 | 87.8/90.8 |
> > > > > | QNLI  | Accuracy                     | 89.2       | 90.1 |
> > > > > | MRPC  | F1/Accuracy                  | 83.5/78.5  |  87.5/83.1         |
> > > > > | MNLI  | Matched acc.                 | 82.6 | 84.0  |
> > > > > | CoLA  | Matthews corr                | 52.15       |  56.49          |
> > > > >
> > > > >
> > > > >
> > > > > Best wishes!
> > > > >
> > > > > Authors of paper 944

---

> > > > > > ### Comment · Reviewer_jXyx · 2021-11-26
> > > > > > **Thanks for the new experiments**
> > > > > >
> > > > > > Thank you for the clarifications and the new experiments. I will raise my score to 6. However, I recommend better tuning of AdamW for the GLUE experiments, because the baselines are rather weak, i.e. not all tasks match the results in Table 1 of the BERT paper (https://arxiv.org/pdf/1810.04805.pdf). I would also suggest that the authors add the new experiments in the Appendix and fix the typos in the blue text, e.g. missing space before citation and missing closing parenthesis. Also, in the appendix when you discuss the interpretability of the equation from P_1, I think you mean "asihn" instead of "sihn".
> > > > > >
> > > > > > Finally, I would like to note that the paper seems acceptable only after extensive experimental details were made available in the Appendix and the blue text. It's unfortunate these details were missing originally, but I am glad they are available now :).

---

> > > > > > > ### Author Response · Authors · 2021-11-26
> > > > > > > **Thank you**
> > > > > > >
> > > > > > > Dear reviewer jXyx:
> > > > > > >
> > > > > > > We appreciate your carefulness and endorsement! We have fixed these typos locally.
> > > > > > >
> > > > > > > Best,
> > > > > > >
> > > > > > > Authors of paper 944

---

### Author Response · Authors · 2021-11-18
**General response from authors to the reviewers and AC**

Dear all reviewers and AC:

We feel very excited about the plentiful feedbacks from all the reviewers. We highly appreciate all reviewer’s inputs to make our work clear and stronger, and the reviewer's acknowledgement of the theoretical novelty and interest in this work.

Being an “ambitious paper” (quoting jXyx), we tried hard to pack a rich set of algorithmic and experimental results, all into one submission. Limited by the space, we had to compress a bit which seems to cause the many clarification questions from the reviewers. We believe that in our point-to-point responses, they have been fully addressed.

In the revised PDF, we have improved the presentation and corrected several typos.  We have added a new appendix section to cover many experimental details, and have re-organized our answers below into the new appendix.

We are very thankful to see the strong enthusiasm from all reviewers towards our proposed approach. As authors, it feels truly great to receive so many constructive, thoughtful and detailed suggestions or even critiques, which all help improve this paper much further. We will be more than happy to address any additional questions.

Authors of Paper 944

---

### Public Comment · ~Jun_Shu1 · 2022-02-08
**Related work suggestion**

Hi authors,

Thanks for this interesting work for learning to optimize (L2O) by defining the optimizer symbolic space and distilling a numeric L2O to its symbolic counterpart. I'd like to point a (possibly) missing related work, which may be worthwhile to be discussed in the paper.

The Meta-LR-Schedule-Net (MLR-SNet, avariable at https://arxiv.org/pdf/2007.14546.pdf) attempts to learn an adaptive learning-rate-schedule for SGD and its variants. The proposed MLR-SNet is able to be generally applied to different DNN training problems, e.g., image and text classification problems. And the meta-learned LR schedules have similar tendency as specifically pre-defined ones.  Meanwhile, we can transfer our meta-learned MLR-SNet to query tasks like different training epochs, network architectures, data modalities, dataset sizes from the training ones, and achieve comparable or even better performance compared with hand-designed LR schedules specifically designed for the query tasks.  We can help train ResNet-50 on ImageNet dataset and achieve the similar performance with SOTA hand-designed method.

I believe that citing MLR-SNet won't devalue yours at all. If you want, I will be happy to discuss further.

Thank you!

---

### Decision · Program_Chairs · 2022-01-20

**Decision:**

Accept (Poster)

**Comment:**

The paper proposes a method for learning to optimize (L2O) by distilling a numerical L2O optimization rule into a simple mathematical rule, mathematical equation, using special-purpose student learning algorithm.  The motivation for using a symbolic distillation is to provide interpretability and scalability of the trained optimizers.

Pros
 - The paper addresses an important problem (better understanding learned optimizers).
 - The experiments give good evidence that learned black-box optimizers can be mapped to mathematical rules.

Cons
 - The symbolic regression student learning algorithm, and many details of the experimentation, remain hard to understand.

Overall, after discussion, the paper was viewed as a solid contribution by the reviews, with some slight disagreement about whether the clarity was sufficient after revisions.  However, I believe that the main points of the paper and the general approach are quite clear, and that the details of the experiments and student learner are sufficiently well-explained for other researchers to build on, at least given the appendix and the supplementary material.